# A Preliminary Study on the Use of HD-sEMG for the Functional Imaging of Equine Superficial Muscle Activation during Dynamic Mobilization Exercises

**DOI:** 10.3390/ani12060785

**Published:** 2022-03-20

**Authors:** Fiorenza Gamucci, Marcello Pallante, Sybille Molle, Enrico Merlo, Andrea Bertuglia

**Affiliations:** 1Dipartimento di Scienze Veterinarie, Università di Torino, 10095 Grugliasco, Italy; fiorenza.gamucci@edu.unito.it (F.G.); marcello.pallante@unito.it (M.P.); 2Independent Researcher, DVM, Private Practice, 28100 Novara, Italy; sybillemolle@gmail.com; 3OT Bioelettronica S.r.l., 10134 Torino, Italy; e.merlo@otbioelettronica.it

**Keywords:** electromyography, objective outcome measurement, dynamic mobilization exercises, rehabilitation, horses

## Abstract

**Simple Summary:**

An evidence-based approaches to assess the effect of rehabilitation protocols is currently under investigation in animal physiotherapy. Research aimed to identify objective outcome measurements are of major importance in this field. Functional muscle assessment remains challenging during the application of specific exercises to activate core muscles in horses. This study introduces for the first time the use of bi-dimensional High-Density Surface Electromyography (HD-sEMG) in horse as a non-invasive tool that provides a functional imaging of superficial skeletal muscle activation, and contraction. Using a 32 and/or 64 surface EMG electrode grids positioned on two superficial core muscles, a colour scale image obtained from recorded signals allowed visual assessment of muscle activation. The exercises used to test this technology were the destabilization exercise, the “chin-to-lateral” bending and the “chin-to-knees” flexion exercise. Our results support the use of HD-sEMG in equine physiotherapy to obtain an immediate biofeedback of muscle activation and objectively tracking the physiotherapy treatments.

**Abstract:**

Superficial skeletal muscle activation is associated with an electric activity. Bidimensional High-Density Surface Electromyography (HD-sEMG) is a non-invasive technique that uses a grid of equally spaced electrodes applied on the skin surface to detect and portray superficial skeletal muscle activation. The goal of the study was to evaluate the feasibility of HD-sEMG to detect electrical activation of skeletal muscle and its application during rehabilitation exercises in horses. To fulfil this aim, activation of the superficial descending pectoral and external abdominal oblique core muscles were measured using HD-sEMG technology during dynamic mobilization exercises to induce lateral bending and flexion/extension tasks of the trunk. Masseter muscle was instrumented during mastication as a control condition. A 64 surface EMG channel wireless system was used with a single 64 electrode grid or a pair of 32 electrode grids. HD-sEMG provided unique information on the muscular activation onset, duration, and offset, along each motor task, and permitting inferences about the motor control strategy actuated by the central nervous system. Signals were further processed to obtain firing frequencies of few motor-neurons. Estimation of electromyographic amplitude and spectral parameters allowed detecting the onset of muscular fatigue during the motor tasks performed. HD-sEMG allows the assessment of muscular activation in horses performing specific motor tasks, supporting its future application in clinical and research settings.

## 1. Introduction

Evidence-based approaches and objective outcome measurements in equine rehabilitation are of major importance, so the development of a new tool for the objective assessment of muscle outcomes is highly valuable [1]. Non-invasive surface electromyography (sEMG) technology to assess muscle function has recently been proposed, thanks to novel wireless technology, for investigation of the neuromuscular system [2,3]. 

sEMG is widely used to understand the physiology of muscle activation when performing specific motor tasks [3,4]. Most studies have used bipolar sEMG electrodes when investigating large superficial muscles in animals [5]. Based on human studies, a single bipolar sEMG can only sample those motor units with fibers located 10–12 mm to the electrodes [6]. Therefore, bipolar sEMG provides only a local sampling of muscle activity in muscles that contain hundreds to thousands of fibers per motor unit [6]. The accuracy of muscle activation estimates based on bipolar sEMG depends on muscle size and electrode position [6]. A growing number of works in the literature report regional patterns of muscle activation [6]. Array or multielectrode bidimensional grids covering a larger part of the muscle surface, on the other hand, are capable of recording motor unit potentials in different parts of a large muscle, increasing the amount of information extracted from sEMG [6]. This technology, referred to as high-density sEMG (HD-sEMG) provides additional physiological information which cannot be extracted with bipolar sEMG. In fact, HD-sEMG can supply details about the anatomy of a muscle (such as the position of the innervation zone and direction of the muscle fibers), the muscle fiber conduction velocity (which reflects the type of contracting fibers), the muscular fatigue, the distribution of the muscle activity represented with a colored map and, by means of an algorithm, the central nervous system strategy through the motor neurons firings [6,7]. To date, no studies have employed HD-sEMG on equine subjects, so the feasibility of this technology as an objective measure of equine muscle activation remains unknown.

In animal sciences, the use of sEMG is considerably less often reported compared to human literature. A recent systematic review of sEMG in veterinary medicine identified 38 papers, out of which 28 investigated muscle function in sport horses [8]. Of these, 23 studies analyzed specific muscles during locomotion [5,9,10,11,12]. Within five studies collecting sEMG data during activities other than locomotion, one investigated chewing, one investigated neck mobilization exercises and three studies investigated myotatic reflexes during active trunk flexion induced by a blunt object running over the hindquarters [8,13]. Three other studies evaluated muscle activity during jumping [14]. Guidelines on best practice for bipolar sEMG signal processing have been recently described [15,16,17], but methodological limitations still exist in the interpretation of the signals. None of the published studies in the veterinary literature used bidimensional grids with multiple array electrodes to record electromyographic signals. 

Core musculature in horse is essential for the maintenance of the posture and the transfer of locomotor forces generated by the limbs to the vertebral column [18,19]. In veterinary rehabilitation, dynamic mobilization exercises are well described and used for core muscle activation. Such exercises consist of executing a controlled movement pattern that recruits both long mobilizing muscles and deep stabilizing muscles to round or bend the neck and back [20]. They include rounding and lateral bending, where many muscles are recruited including the epaxial, abdominal, pelvic, and pectoral muscles. Among these exercises, the sustained chin-to-flank lateral bending, the chin-between-knees flexion and destabilization exercises are commonly used for core muscle activation. The chin-to-flank lateral bending exercises move the head and neck laterally and are performed to the right and left side. This exercise elicits most of the bending at the cervicothoracic junction with some flexion and lateral bending of the thoracic spine. The chin-between-knees flexion exercises stimulates the rounding of the base of the neck and cranial thoracic region with a different stimulation of the muscles of the neck and trunk in the different positions.

Balancing exercises aim at stimulating the recruitment of various components of the core musculature by perturbing the horse’s balance in different directions. Repetition of those exercises leads to improvement in balance and stability [18]. 

Destabilization exercise, where a forelimb is raised/picked-up and pressure is applied to the sternum or at the point of shoulder to shift the weight caudally, results in the activation of the muscles of the thoracic sling and pelvic stabilizers. Variation of this exercise could be performed exerting a direct pressure on the withers, promoting a lateral shift of the horse’ weight.

The efficacy of dynamic mobilization exercises is based on the evaluation of its effects measuring muscle dimension by ultrasonography in two clinical trials [21,22].

The main purpose of our study is to identify a direct biofeedback of muscle activation in two superficial core muscles in horses using HD-sEMG technology and to demonstrate that dynamic mobilization exercises can produce an electromyographic trace in those muscles. Secondarily, we hypothesized that HD-sEMG, during extended muscle activation, can identify the onset of muscle fatigue, associated with a reduction in muscle force and an increase in the motor units’ recruitment to maintain muscular activity level.

## 2. Materials and Methods

### 2.1. Subjects

In total, 10 healthy horses of different breeds and ages, performing specific predetermined motor tasks, were enrolled in this study over three different trials (Table 1). Horses performing trials were not sedated and data collection took place in their everyday environment. 

### 2.2. Equipment

An HD-sEMG system developed by OT Bioelettronica s.r.l (Sessantaquattro, OT Bioelettronica, Torino, Italy) was employed (Figure 1). This wearable device can detect up to 64 sEMG signals simultaneously from electrode grids and operates wirelessly (Wi-Fi based). Amplified signals at bandwidth 10–500 Hz were registered and converted in digital values with a resolution of 16 bits and a sampling frequency of 2000 Hz. The data acquired can be transferred to a PC for real-time display and storing. Data can also be recorded locally on a MicroSD card using the device as a Datalogger.

Two different grid types were employed, based on the size of the instrumented muscle: 32 electrodes (8 rows and 4 columns) and 64 electrodes (8 rows and 8 columns). Electrode grids are flexible, resistant, and semi-disposable, with a 10 mm inter-electrode distance (IED). The grids are applied on the skin by means of a double adhesive foam with 3 mm-diameter holes corresponding to the electrode positions. The holes are filled with conductive cream, creating the contact between the skin and the gold-coated electrodes.

### 2.3. Muscles Instrumentation and Motor Tasks

Instrumented muscles were superficially positioned with limited superimposition with other muscle bellies. Two core muscles have been considered due to the possibility of easily reproduce a motor task to activate them: the superficial descending pectoral (SDP) and the external abdominal oblique (EAO). 

A further muscle, the masseter muscle, was instrumented during spontaneous chewing, because activation of masticatory muscles is certainly obtained offering to the horse a reward (good quality hay) and easily recorded due to the superficial position. The masseter muscle was considered in this study as a positive control to demonstrate the feasibility of the system in horse. 

All exercises were performed by one of the authors (S.M.) with specific training in equine rehabilitation.

The EAO muscle was investigated in horses previously trained to perform dynamic mobilisation exercises. Only a portion of this muscle was studied because of its easy access and flat surface, corresponding to the proximal insertion on the 16th rib, directly under the skin of the paralumbar fossa. Two 32-electrode grids were positioned at the level of the 16th rib, referring to the location reported in a previous study [12]. Both the left and right EAO muscles were instrumented simultaneously. The grid was positioned parallel to the direction of the muscle fibers, approximately 10° in the proximocranial–distocaudal direction. The motor tasks performed for EAO activation were elicited using a carrot as a reward: a sustained “chin-to-flank” lateral bending and a “chin-between-knees” flexion.

The SDP muscle was investigated due to its superficial position and its role in stabilizing the thoracic girdle [20]. Two different 32-electrodes grid were placed parallel to the direction of the muscle fibers. The motor task for the SDP activation consisted of destabilization exercises of the forehand, realized by picking up one forelimb and applying a direct pressure on the withers, promoting a lateral shift of the weight. This position was held for 20 s then the pressure was released [18]. Thoracic girdle activation was further assessed during forward and backward strides, when horses were encouraged to move one step forward and backward.

The masseter muscle was instrumented with a single 64-electrode grid positioned over the masseter region, using reference landmarks previously described in a surface electromyography study using bipolar surface electrodes [13]. The motor task used for the masseter muscle activation was mastication, obtained by offering good quality hay to the horse. 

### 2.4. HD-sEMG Acquisition

Skin preparation is essential for an optimal recording of low noise-level electromyography signals using multiple-array electrode grids [16]. Skin preparation was performed to assure the complete removal of hairs and flaky skin layers for proper electrode adhesion of the skin–electrode interface [17]. Skin was clipped and shaved by hand. After washing with a neutral soap solution, it was further cleaned with an abrasive cream (Spes Medica s.r.l., Genova, Italy). Grids were coupled with a specific double-sided adhesive foam pad punctured in correspondence with each electrode. The foam pads were spread on the free side with a conductive cream (AC Cream, Spes Medica s.r.l., Genova, Italy) to enable optimal contact between the electrodes and the skin surface. Sessantaquattro system was secured to the patient. The zero-voltage reference point was positioned at the level of a bony prominence based on the instrumented muscle (tuber coxae for EAO, the supraglenoid tuberosity of the scapula for SDP and the temporal bone for the masseter muscle). Each motor task was repeated until a clean electromyography signal was recorded without mute channels recording or skin-to-electrode motion artifacts. This means that a coherent appearance of the myographic signals need to be detected by the examiner during motor task recording to acquire the record. 

### 2.5. Data Acquisition, Postprocessing and HD-sEMG Analysis

Monopolar electromyographic signals (i.e., the voltage detected from each grid electrode with respect to the reference electrode) acquired during motor tasks were displayed in real time on a laptop screen. The HD-sEMG can be displayed as raw signals or as a colorimetric image representing the electric potential amplitude distribution on the skin, as detected by the electrode grids, over short time epochs (usually 0.5 s). Out of several trials, the better signals were recorded and archived based on subjective assessment of the operator though direct visualisation of the images. Collected data were post-processed with OTBioLab+ software (OTBiolettronica, Torino, Italy) with a band-pass digital filter (20–400 Hz) to eliminate high-frequency noise and low-frequency fluctuation due to artifact movements. Furthermore, the differential signals were estimated as the difference between adjacent electrodes along the grid columns oriented in the muscle fiber direction, to highlight the motor unit action potential (MUAP) propagation and reducing common mode signals and crosstalk. In fact, the differentiation between signals detected from adjacent electrodes introduce a spatial filter that reduces the amplitude of signals that are generated far from the detecting electrodes and appear as a common mode component in the sEMG signals. From the postprocessed signals, a series of images have been generated to allow the user to directly visualize the muscular activation during the specific motor task [17].

Through a plugin named “Decomponi” and based on a blind source separation algorithm, the OTBioLab+ software can decompose the HD-sEMG signal recordings, extracting a subset of the active motor units and generating the spike trains corresponding to the motor unit fires [23,24]. To properly identify the motor unit spike train, the algorithm needs to have at least 20 s of a stable activation with the same motor unit active and firing.

The root mean square (RMS), representing the EMG amplitude over 0.5 s epochs and expressed in μV, was chosen to investigate the amplitude features of electromyographic signals. The mean frequency (MNF) and the median frequency (MDF), representing the frequency content over 0.5 s epochs and expressed in Hz were calculated to examine the spectral features. For each muscle and for each motor task, the RMS, the MNF and the MDF were calculated and exported in a table footnote (Microsoft Excel TM) for the subsequent analysis. This computation allows the user to assess the characteristics of the muscular activation and to detect the onset of muscle fatigue.

To analyse myoelectric manifestations of muscle fatigue and allow comparison between different variables, muscles and subjects, a fatigue plot was generated. Myoelectrical signs of fatigue are identified by an increase in RMS values within time and a decrease in MNF and MDF values. This fatigue indicators’ pattern is related to the reduction of muscle fiber conduction velocity, which decreases more or less rapidly depending on the level of contraction and causing a change in the power spectrum of the sEMG toward the lower frequencies, with a decrement of the mean and median spectral frequencies [2,25].

To determine the mean of the RMS, MNF and the MDF over time, linear regressions has been used. To highlight the trend of the entire time series of raw signals, linear regressions of RMS, MNF and MDF in time were computed using the following formula:Y = α + β × TIME_t + ε_(i,t)(1)

The dependent variable Y is the RMS mean over channels, MNF or the MDF. TIME_t is the time in seconds. ε_(i,t) is the error term. Depending on the model, β captures the marginal effect of an increase of the time (TIME_t) on the RMS, MNF or the MDF, while α captures the trend of the data. The trend of the data was captured and evidenced by the alpha of the regression. To assess the power of the model, the R-square (R2), which ranges from 0 (not significant) to 1 (perfect fit), was considered.

## 3. Results

### 3.1. Horse’ Demography

Horses’ age, fitness and work level were different between subjects.

### 3.2. Raw Signals and Time Features 

The evaluation of the signal acquired along the single channels of each grid provided information on the onset, offset and duration of muscular activation for each motor task. Positive depolarization and repolarization waves were not detected in any of the muscles during the resting phase, due to the absence of muscular activity. Signals were visually and subjectively assessed during motor tasks: ▪Mastication: the masseter muscles’ raw signal shows burst-like muscular activity during mastication (Figure 2). ▪Destabilization exercise and forward/backward stride: SDP muscle presents burst-like muscular activity in both lateral swings with the contralateral limb picked-up (Appendix A) and forward strides (Figure 3). The muscle was not activated during a single one-stride backward. ▪Sustained chin-to-flank lateral bending and chin-between-knees flexion: the instrumented portion of EAO produced a single activation burst during the chin-to-flank lateral bending, only on the flexion side. During the chin-between-knees flexion exercise, a stationary muscular activation was demonstrated in both the right and left EOA muscles (Figure 4). 

Data records presented in the manuscript and accompanying detailed description file are presented as Appendix A. The records contain the raw signals without any further processing. Data are stored in individual folders for each of the subject (OTBioLab+ v1.5.7.2.exe required to open the files). Files are named according to the type of task recorded.

### 3.3. HD-sEMG Features

Raw differential voltage along the muscular fibers’ direction generated color maps of the entire electrical activities of the instrumented muscles. Corresponding color maps were plotted for the different muscles and tasks. Maps are unique feature to visually identify the muscular activation during the assigned task. Each point on the map represents an electrode on the grid, while the arrow indicates the direction of the computed differential. Color scale to generate the maps has been set in the range between −100/−150 microvolts (blue) and +100/+150 microvolts (red), which allows the migrating MUAPs representing the muscular activation through time to be better highlighted. MUAPs have a positive (red) depolarization wave, followed by a negative (blue) repolarization wave (Figure 2, Figure 3 and Figure 4). 

Using a time setting in milliseconds, the surface electrode grids used in this study enable the detection of a so-called electromyographic “image” that is evolving in time similar to a series of a frame, providing a time-varying electrical image of the direction of the MUAP depolarization waves in the muscle (Appendix A). The region comprising the neuromuscular junction in the instrumented muscles, called the motor unit innervation zone (IZ), were identified as a splitting of the depolarization waves, with two waves travelling in opposite directions (Additional file: Figure 1) [26]. The number of motor units recruited can be calculated in recordings of at least 20 s through the Decomponi plugin (Figure 5).

### 3.4. Amplitude and Spectral Features of HD-sEMG

The analysis of spectral features shows a significant increase in the RMS and decreases of the MNF and MDF across all motor tasks. This feature could indicate that all skeletal muscles were subjected to detectable fatigue during proposed sustained exercises (Figure 6).

An increase of the normalized value of RMS and a decrease for the normalized value of MNF and MDF were observed in the right EAO muscle during ipsilateral flexion. The R^2^ value of the models are relatively high for the RMS (Figure 7). Those results show the goodness of fit of the models (second-degree polynomial models could have been used, but for simplicity, we privileged linear models). When comparing the three features’ series variations (RMS, MNF, and MDF), a significant difference was found between RMS, that showed a marked increase and the values of MNF and MDF, that remained close to a steady state. Results obtained from the left EAO during ipsilateral flexion showed a similar increasing distribution in the RMS, while the MNF and MDF remained stationary or increased over time, indicating that the left EAO was not subject to detectable fatigue in the horses used in this study.

During repeated episode of mastication, masseter muscle had similar spectral features to those observed in the right EAO.

SDP muscle was considered only during the left swinging motor task, due to the short duration of the task and the loss of contact between the skin and the electrodes grid during the forward and backward strides. The RMS increased during the muscular activation of SDP muscle, while the MNF and MDF tended to decrease over time (Figure 8).

## 4. Discussion

This study highlights that HD-sEMG in horses enables the assessment and investigation of skeletal muscle activation and its pattern of activity, with the ability to measure muscular fatigue during specific motor tasks. The HD-sEMG uses dense grids of equally spaced flexible surface electrodes, applied to a single muscle to record the myoelectric activity and to extract anatomical and physiological information, at the motor unit level. The output, expressed in terms of voltage changes and referred to the MUAP, represents the bidimensional electrical voltage distribution in a corresponding activation map of the instrumented muscle. Microvoltages registered on the skin above a specific muscle are represented with a false-color scale map set by the operator. The activation maps provide both the intensity and the spatial distribution of the muscular activity under the grid, giving a direct biofeedback to the operator during exercises. The main HD-sEMG features is the time-related capability to investigate the onset and the pattern of activation of a muscle in a bidimensional way. Since raw data can contain spurious electrical activity, they are postprocessed to effectively enable data assessment and to evaluate muscle recruitment and activity [27]. Moreover, the amplitude feature, derived from the RMS, indicates the power of the myoelectrical signal. The spectral features (MNF and MDF) provide information about the muscle activation strength and the signal frequency, enabling, for instance, the identification of the onset of fatigue [2,25]. This study, to the best of our knowledge, is the first in which an HD-sEMG system was successfully applied to horses at the level of selected core muscles. 

Masseter muscle was instrumented in this study because its activation is certainly obtained during mastication cycles. This is simple to realize by feeding a horse with a reward. Masseter muscle was activated during chewing and its spectrum of activity analysis can be helpful to demonstrate the effectiveness of the system that was employed for the first time in horses. 

The activation of SDP muscle was obtained with proper application of destabilization exercises. Moreover, backward stride did not have any effect on the activation of this muscle since the muscle is responsible for limb protraction. 

Both lateral bending and rounding exercises were able to selectively activate the instrumented portion of EAO muscle. This is a useful feature for demonstrating the effect of the sustained chin-to-flank lateral bending and chin-between-knees rounding to activate hypaxial core muscles in horses during the execution of the exercises. 

Only a portion of large EOA muscles has been instrumented in our research because of the limitation imposed by the size of the available electrode grids. A certain degree of approximation has been necessary, because grids size do not fit perfectly with the extension of certain large muscles in horses. Despite this, activation profile defined for a single EOA insertion could be translated to the other muscular insertions of this large abdominal muscle. Findings from this study supported that the proposed exercises can be considered suitable for strength training of the EAO muscle and SDP muscles in horses, with successful activation of core musculature. 

We found similar variations in the electromyographic spectral features in all the muscles analyzed during the proposed sustained motor tasks. We analyzed a spectrum of muscle activation along 10 to 20 s of muscle activation. This caused a rise in the number of motor units recruited along the muscular activation, to maintain the muscular force imposed by the requested exercise. A different effort along the selected motor tasks reflects individual muscular profiles in terms of muscular adaptation and is a unique opportunity to evaluate muscle activation during dynamic mobilization exercises.

The spectrum of information obtained from HD-sEMG is different compared to the information obtained using classic surface or indwelling needle electromyography. The advantage of HD-sEMG application in comparison to traditional electromyography is the ability to obtain a surface electromyography imaging of the active portion of the muscle, to identify fiber length and orientation and to visualize innervation zones of the muscle. Biofeedback for optimal muscle involvement while performing a task is the principal value of this technique, but still require further exploration. Disadvantages of HD-sEMG are related to the limitations encountered in this study. Skin elasticity and sweating under the grids made it difficult to have long time adhesion of the electrodes to the equine skin resulting, in poor electrical contact over time. Skin preparation was revealed to be fundamental for proper data acquisition, especially in terms of contact between the skin and electrode grids for recording high-quality electromyograms. Motor tasks in our study have been studied without kinematic data acquired simultaneously to the sEMG acquisition, and it was obtained in a limited cohort of horses. Lastly, most research on electromyographic studies are executed during a sustained muscular contraction, at a predetermined percentage of the MVC during selected motor tasks. This type of analysis is not feasible in animals due to the impossibility of eliciting an MVC during an incremental muscular activity. Therefore, our motor tasks could not meet the same criteria used in human studies [3].

All the skeletal muscles of the horses studied in this paper presented similar muscular activation pattern; however, the same muscles present different fatigue trends. The sEMG data interpretation in relation to fatigue remains challenging because the signals could be affected by many different biases [28]. For instance, the subject’s muscular characteristics and the training level of the horse can influence the sEMG profile concerning the arising of muscular fatigue. Moreover, in humans, greater fluctuations in the sEMG features’ analysis are shown, and therefore also in fatigue, during voluntary activation [25]. Concerning the fatigue onset, in our study the amplitude and spectral features were analyzed based on algorithms used in the human literature. Spectral features are computed to investigate the onset of fatigue, but in this preliminary study there are insufficient repetitions and inconstant data to be consistent on this topic. Further studies are needed to better evaluate the characteristics of muscular activation in horses, improving the reliability of the motor tasks, and better correlate the onset of muscle fatigue with specific motor tasks. The absence of sustained isometric voluntary contractions could make the comparison of muscular fatigue difficult between different subjects [27]. Electromyographic features showed a greater myoelectrical manifestation of muscle fatigue during supramaximal stimulations or during MVC. Williams et al. (2013) [29] found no significant differences in the mean frequency of the signals within the studied group of horses, however muscular fatigue was evidenced by the variation of the frequencies, suggesting decreased muscle effort after training, but no clear relationship was found between fitness level and fatigue parameters. Conversely, Colborne et al. (2001) and Cheung et al. (2018) [30,31] found that spectral features decreased through fatigue onset. Furthermore, it is now evident that comparisons of electromyographic features between muscles, subjects or motor tasks are highly critical and it would be more reliable to compare this data at the individual level [32]. Variations in the amplitude of muscle activation can be attributed to an increased recruitment of motor units or firing rate [25]. An amplitude increase can also be representative of a greater workload associated with the muscle shortening during concentric contractions, rather than during eccentric work. Additionally, a high-amplitude signal can be explained as an increase in fast-twitch fiber recruitment, or as a high firing rate in slow-twitch fibers, due to the onset of muscular fatigue. In the horse, the superficial compartment of core muscles is composed predominantly of fast-twitch fibers, facilitating short duration and rapid force production [6]. Meanwhile, the deeper compartment of skeletal muscles is manly composed of slow-twitch fibers [2,6]. Therefore, our electromyographic results could represent the recruitment of more fast-twitch fiber activity rather than the summation of muscle activity. The instrumented horses in this study are all taking part in recreational riding, without specific training. Data interpretation on this topic after specific training requires further research. Indeed, the preliminary data obtained through this work investigating specific muscle functionality offers a guideline for the proper acquisition of data. 

We demonstrated that the HD-sEMG is an effective functional imaging technique to obtain direct biofeedback of superficial skeletal muscles electrophysiology in horses and it can be introduced in equine research. A couple of barriers to the widespread application of sEMG in practice have been recently identified in the human (cultural, educational, and technical) context [33] and the same could be a potential problem in equine medicine. For future research using HD-sEMG, researchers should give particular attention to the study design of the trials and to the data analysis and interpretation. There is a similar need in the human field of generated multicentric collaboration projects such as the SENIAM project (Surface Electromyography for Non-Invasive Assessment of Muscles) collecting the experience of many groups working in this field, allowing for an increase in consistency of data and acquisitions [34,35]. A similar project in horses might be useful to promote consistency in sensor placement, data collection, data analysis and results interpretation. A further development field of this work could involve an implementation of HD-sEMG data in kinematic, increasing the knowledge on muscle recruitment and coordination during movement and providing additional investigation of the correct muscle involvement while performing a task and the onset of fatigue.

## 5. Conclusions

This study is a first step of applying a new functional technology capable of demonstrating core muscular activation in horses. Thanks to its wearable and non-invasive characteristics, the HD-sEMG technique can offer new opportunities in clinical research to detect muscular activation during specific motor tasks in horses.

## Figures and Tables

**Figure 1 animals-12-00785-f001:**
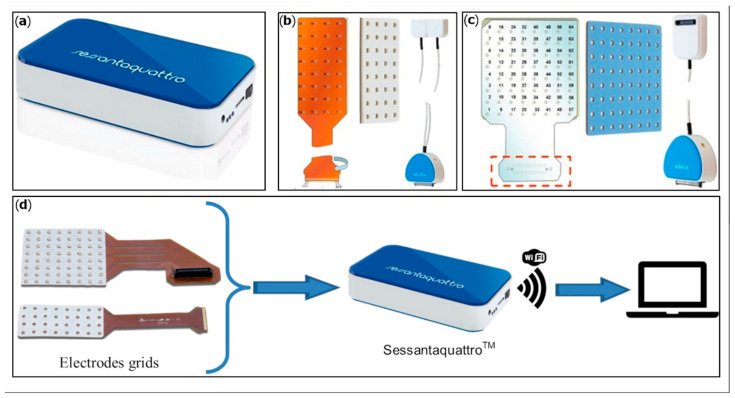
(**a**) Sessantaquattro main unit; (**b**) 32-electrode grid, corresponding biadhesive foam pad and adapter; (**c**) 64-electrode grid, corresponding biadhesive foam pad and adapter; (**d**) grids related to the Sessantaquattro main unit data acquired. The Sessantaquattro unit can transfer electromyographic signals wirelessly to an external laptop.

**Figure 2 animals-12-00785-f002:**
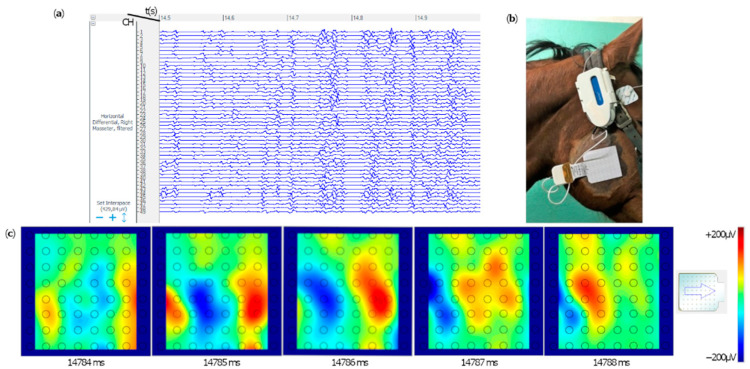
Extract of single differentials of the electromyographic signals in the right masseter muscle, instrumented with a 64-electrode grid, during mastication (Horse ID 5). Masseter activation was observed. Time flow expressed in seconds is indicated on the top (timeframe between 14.5 to 15 s). Each horizontal line represents the signal difference between two consecutive channels in the corresponding electrodes column (**a**). The position of the grid over the masseter muscle is shown (**b**). From left to right five consecutive color maps show muscular activation (**c**). The color scale on the right side is the color map calibration (±200 μV).

**Figure 3 animals-12-00785-f003:**
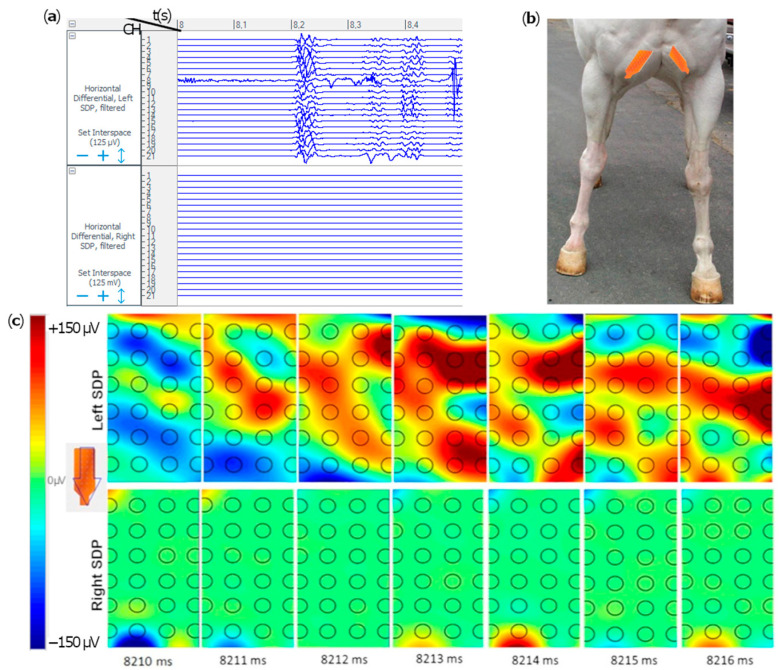
Extract of single differentials of the electromyographic signals in the superficial descending pectoral (SDP) muscles in a single horse. Some interference attributed by movement artifact are present in channel 8 (**a**). The position of the two electrode grids on the skin is shown in (**b**). Frame sequences (7 milliseconds) show the activation map of the SDP muscles during left forelimb forward stride. The time is reported below the frames sequence (from 8 to 8.5 s). Color maps shown are relative to the left and right SDP activity (**c**). The color scale on the left side is the color map calibration (−150 and +150 μV). The orientation grid with the arrow indicates the direction of the computed differential. In the left SPD, two positive depolarization waves (red waves) are clearly detectable, followed by a negative repolarization wave (in blue) traveling distally along the muscle. Conversely, the right SPD muscle shows no activation in the same epoch.

**Figure 4 animals-12-00785-f004:**
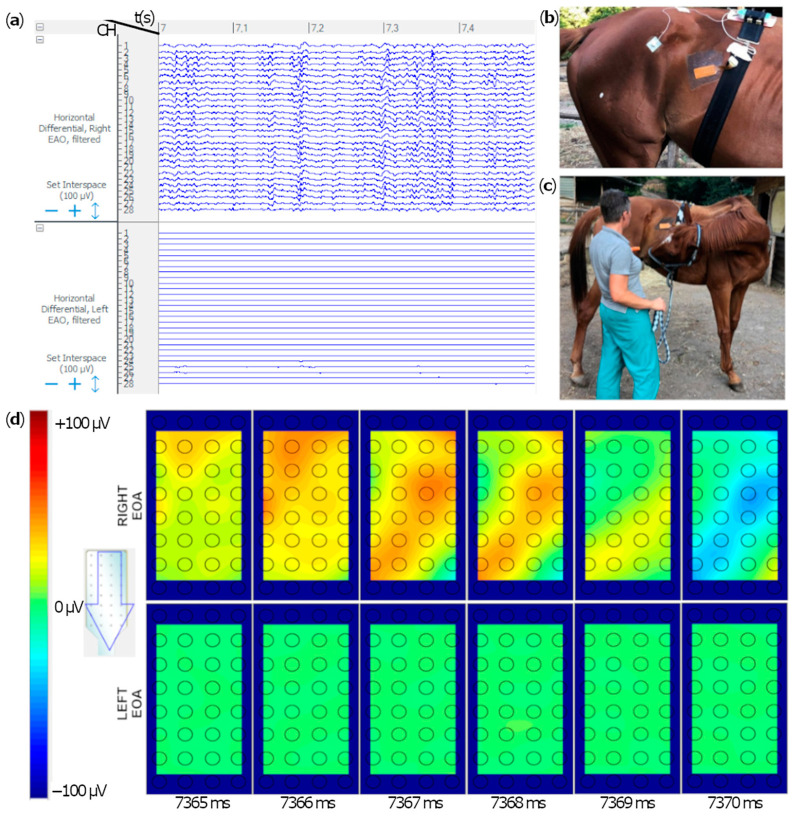
Extract of the single differentials electromyographic signals in the right and left EAO muscles, instrumented simultaneously with 32-electrode grids, for 10 s in a horse. Time flow expressed in seconds is indicated on the top (from 7 to 7.5 s). Each horizontal line represents the signal difference between two consecutive channels in the corresponding electrodes column (**a**). Electrode grids’ positions and orientations are shown in panels (**b**). Horse was given a carrot to obtain a sustained chin-to-lateral flexion (**c**). Activation maps of an instrumented portion of the right External Abdominal Oblique (EAO) muscles in six consecutive color maps, from left to right (**d**). Color scale of the image is set between −100 μV and + 100 μV. Maps in the upper line represent muscular activation of the right EAO, whereas maps in the lower line represent the left EAO activation. Two action potentials (red waves) are travelling peripherally. The red waves split indicates the innervation zone (IZ) in the muscle, followed by a repolarization wave (in blue), as shown in (**d**).

**Figure 5 animals-12-00785-f005:**
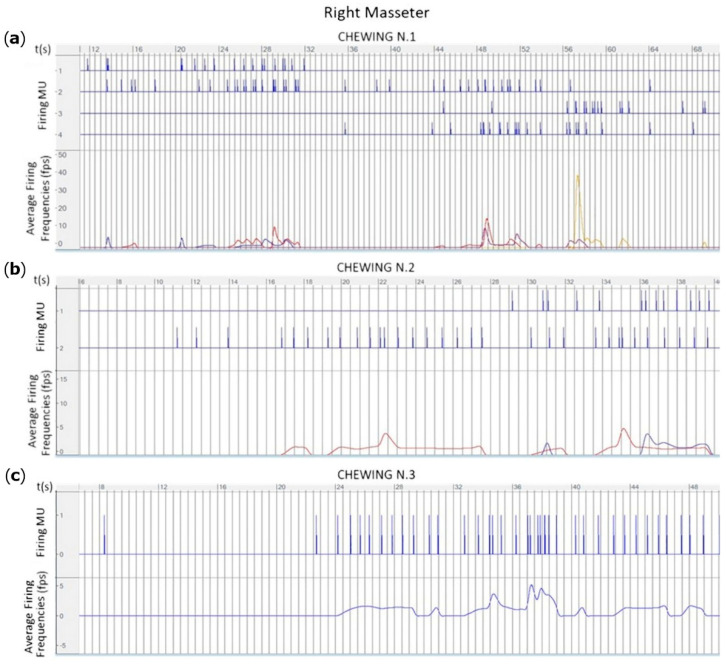
The output of the Decomponi plugin in terms of number of motor units activated during three cycles of mastication in three different horses (**a**–**c**) in the right masseter muscle are shown. The upper part of each graph shows the firings motor units extracted from the signal and their firing instants in function of the time. The lower part of each graphic shows the average firing frequencies (fps) of each motor unit (coloured lines) in function of the time.

**Figure 6 animals-12-00785-f006:**
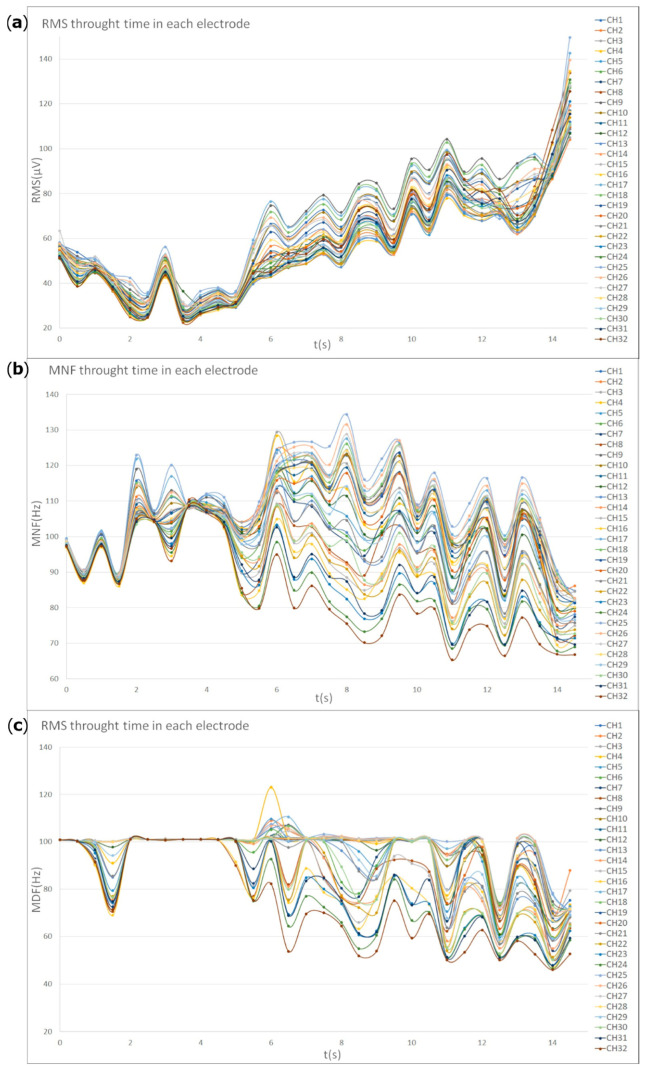
Root mean square, mean frequency and median frequency distribution graphs of the 32-electrode grid positioned on the right EAO, during a sustained right chin-to-lateral flexion. The acquisition time was around 14 s. On the x-axis the time is expressed in seconds, while on the y-axis the root mean square (**a**), the mean frequency (**b**), or the median frequency (**c**) are shown. Each coloured line in the graph represents a different electromyographic channel of the bidimensional grid. A single horse is shown.

**Figure 7 animals-12-00785-f007:**
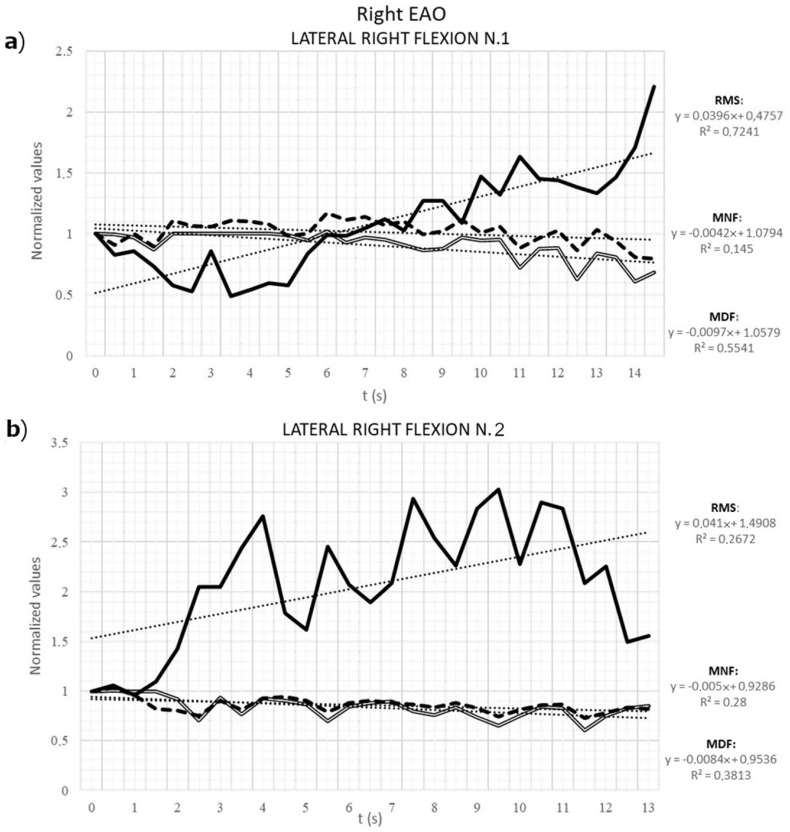
Normalized means of the root mean square (RMS, black continuous line), mean frequency (MNF, black dashed line) and median frequency (MDF, contoured black line) of the 32-electrode grid positioned on the external abdominal oblique muscle during two repetition of right chin-to-lateral flexions motor task. The graph (**a**) corresponds to the first repetition, while the graph (**b**) corresponds to the second repetition. The regression lines (dotted lines) enable to capture the trend of RMS, MNF and MDF during time and provide the R^2^ explanatory model power.

**Figure 8 animals-12-00785-f008:**
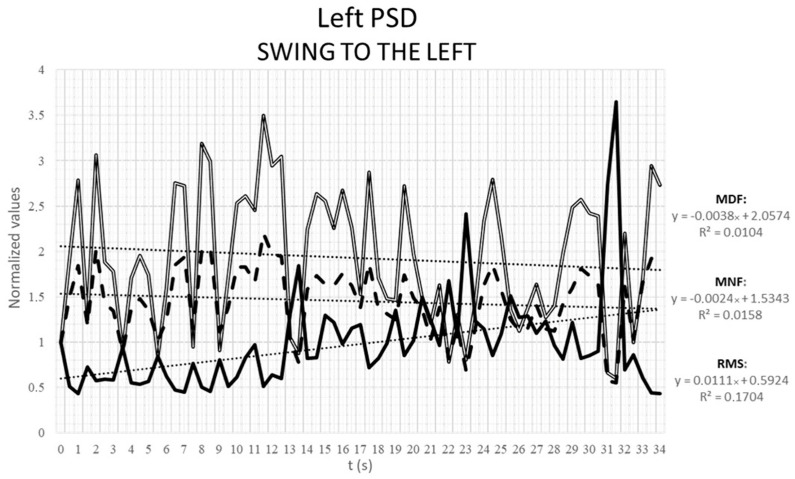
Normalized means of the root mean square (RMS, black continuous line), mean frequency (MNF, black dashed line) and median frequency (MDF, white line) of the 32-electrode grid positioned on the left superficial descending pectoral muscle during a destabilization exercise (swing to the left).

**Table 1 animals-12-00785-t001:** Horses’ demographic characteristic.

Horse ID	Age	Sex	Breed	Weight	Type of Work
1	18	F	Warmblood	550	Recreational riding
2	12	F	Arab	460	Recreational riding
3	11	G	TBR cross	470	Recreational riding
4	10	G	Warmblood	530	Recreational riding
5	8	F	Quarter horse	475	Recreational riding
6	15	G	Warmblood	480	Recreational riding
7	12	F	Standardbred	490	Mare
8	11	G	Warmblood	520	Recreational riding
9	19	F	Arabian cross	460	Recreational riding
10	8	F	TBR cross	540	Recreational riding

## Data Availability

Not applicable.

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
