# Peer review of "A Preliminary Study on the Use of HD-sEMG for the Functional Imaging of Equine Superficial Muscle Activation during Dynamic Mobilization Exercises"

_animals, 2022, doi:10.3390/ani12060785_

Round 1

Reviewer 1 Report

This is an interesting study that evaluates the feasibility of HD-sEMG for objectively monitoring/measuring muscle activation in equine subjects, with a specific focus for its use in equine rehabilitation. Indeed, this study is required as a foundation for evaluating the usefulness of this technology for horses and I would really like to see this work published. However, in it's current form, I felt that the manuscript lacks a clear focus, flow, and structure. As an experienced reader/researcher in this field, it is clear to me that the author’s main aim is to evaluate the feasibility of HD-sEMG for use in horses, but I do not feel that this will always be clear to the readership. A number of variables are employed to fulfil the aims and these are calcuated from many tasks, some of which seem very unrelated (i.e. mastication and dynamic mobilisation exercises used for equine rehabilitation/evaluation). The manuscript is missing quite a lot of information which is required to ensure its reproducibility and reading through it generally raised more questions than answers for me (some of which were answered much later in the manuscript). As such, I feel that this manuscript requires major revisions, where the authors can focus on presenting the reader with a clear aim, a clear justification for hypotheses and methods and a methodology section that ensures full transparency for reproducibility. Please see below an extensive list of my concerns/feedback for improvement. I hope that these are helpful and can ensure that this work is published. 

Introduction: Sentence structure and grammar should be reviewed throughout the introduction section. I found some sentences were difficult to follow due to grammatical errors, missing words, or run-on sentences. For example: the explanation of HD-sEMG.

The flow of the introduction section could be improved, specifically the author’s presentation of the background/gaps in knowledge for the implementation of HD-sEMG, how this differs from bipolar EMG (commonly used in equine studies), the lack of standardisation in the field of equine sEMG (the authors have missed some references here from St. George et al., who describe the lack of standardisation and have conducted research on this), applications for sEMG within equine rehabilitation. As it stands, each of these issues are raised to the reader, but there is a lack of flow, which links the ideas together and thus it may be difficult for the reader to understand how this impacts the current study’s aims and objectives/why this study is necessary. The authors could consider re-arranging some of the paragraphs so that the readability of the section is improved. My suggestions for this would be to first introduce the gaps in knowledge for objective monitoring/measurement techniques in equine rehabilitation (lines 77 – 87), then introduce the reader to how sEMG technology can be employed to fulfil the need for objective measures, followed by a review of current literature that has employed sEMG in horses, highlighting the lack of research that has employed HD-sEMG in horses (and how this differs from sEMG) and our current gaps in knowledge for standardising equine sEMG acquisition, processing and analysis. This can then lead to the aims and objectives of the study and how these will be achieved. I think that this would greatly improve the section.

Lines 93 – 100: This paragraph requires some attention and improvement. The aim is vague and does not describe the selected motor tasks or mention the dynamic mobilisation exercises (line 96 – 97) that will be used to fulfil it. There are a multitude of dynamic mobilisation exercises, which target different areas/movements in horses, but these are not described in detail anywhere in the introduction (apart from a brief mention in lines 82 – 85). Further, a reference referring to their description is not provided to the reader, apart from [9], which is a book and the reader cannot be expected to read an entire book to access this information. I would encourage the authors to provide more detail on the specific exercises/muscles that they are focussing on in this study and how they are performed, earlier in the introduction section. Further, are the authors not trying to evaluate the feasibility of this technology for use in horses/equine rehabilitation? If so, then this would add clarity to the study aim. The hypotheses are not clearly related to what has been described earlier in the introduction section and the aim of the study. With respect, the hypotheses seem to come up at random and I feel that this could be mitigated by ensuring that the reader is provided with more information within the introduction section that relates to each hypothesis (i.e. ability of HD-sEMG to measure muscle fatigue and dynamic mobilisation exercises in horses). 

Materials and Methods:

Line 103-106: the authors should provide a description of the metrics for the sample (i.e. horse age, breed, sex, height, weight, etc). Further information on each horse's fitness and work level is required. These are all important information in relation to the type of data collected and to ensure repeatability of the work.

Section 2.3: Here I can appreciate that the authors have now provided further information about the dynamic mobilisation tasks that were performed in the study, but I would prefer it if (at least) some of this information were to appear earlier in the manuscript, as well as in this section. This would mitigate the questions that arise for the reader and make them aware of the methods that will be used to fulfil the aim of the study. It’s still unclear to me how mastication is used as a control task within the study. What do these muscles/chewing task control against? The paragraph consisting of lines 132 – 135 could be removed or the authors could consider informing the reader that each of the tasks are described in detail in the proceeding paragraphs. Please include additional references for the justification for muscles/tasks studied in lines 126 – 131 and throughout this section. To ensure reproducibility could you provide further information for whether EAO activation tasks were elicited using a reward (i.e. carrot stretches)? SDP electrode grid location is missing in the text and not included in Figure 1.  Are SDP forward/backward tasks just one step in walk? Or consistent forward steps and backward steps? How many repetitions were required/collected for each horse/task?

Lines 172 – 174: This sentence is interesting. If these tasks have not yet been recorded using sEMG/HD-sEMG how did the authors define “nonsense” amplitudes or patterns? Surely, high-pass filtering protocols could be employed to attenuate low-frequency noise sources from movement artefacts? How many attempts/task repetitions were required for each horse to obtain quality signals? More information required here. 

Section 2.5: This section raises a lot of questions for me and areas where I think further information should be provided to ensure reproducibility of the work. Currently there is not enough information for me to reproduce the post-processing and analysis: These include:

  • Line 180: how were “optimal signals” defined? What was the “defined study period”?
  • Line 181: I would argue that this band-pass filter would attenuate more than just environmental interferences. Movement artefact, cable motion artefact, etc?
  • Lines 184 – 187: the authors could consider referring the reader to some of the existing figures here, to illustrate the different methods/modes for analysis.
  • Line 187 and 195: milli volts or micro volts? Micro volts presented throughout the remainder of manuscript.
  • Line 188: please describe the “further filtering” that was conducted to attenuate noise sources. 
  • 189: Further information required here. Was the method described in [20] employed? This is not clear.
  • Line 191: “and based on an algorithm, described in [21]”?
  • Line 192: “can decompose at least 20 seconds”
  • Line 195: does this paragraph refer to the single differential signals? What was the time domain for each of these variables (RMS, MNF, MDF), so how were the signals analysed in relation to time/the dynamic mobilisation tasks? How can RMS, MNF and MDF be applied to detect the onset of muscle fatigue (shift in median frequency?). Further explanation, with appropriate referencing required here.
  • Line 202: again, here the author’s raise questions that are then addressed in later paragraphs (here they address my questions above about the onset of muscle fatigue). To correct this, combine paragraphs 195 – 206 and explain cohesively.
  • Line 204: what reference values were used for normalisation? The reader should not have to read an entire referenced work to obtain information that can easily be included in the text. Please include this information. 
  • Line 207 – 208: remove this sentence. Again, it just raises questions about whether the authors actually used correlation analysis, which is then answered in the paragraph below (line 209 – 210, which should be combined into one paragraph).
  • Can the authors realistically expect the tasks that are performed to induce fatigue? If not, why study this?

Results:

Line 220 – 221: the authors do not provide any explanation within the methods section about how onset, offset and activity duration were to be assessed. It appears that this was just visually/subjectively assessed, and if so, this should be clearly stated.

Line 222: “Masseter muscles raw signal present burst-like muscular activity during the chewing cycles” – this does not make sense to me. I do not know what the authors are trying to describe. What is the chewing cycle and how many bursts were observed within this cycle? More information required, as it is not a clear description of muscle activation. Figure 2 is very small and thus it is difficult to interpret such a large amount of information. How many chewing cycles are shown here over the approx. 30 second period? How were the epochs of muscular activation differentiated from the other clear (albeit shorter/lower amplitude) bursts of activation that occur throughout the cycle? Are these data from one representative horse? Further information required in the figure description.

Lines 224 – 233: The authors could consider providing figures for these tasks/exercises, as was done for mastication in Figure 2. Again, the explanations of muscle activation remain a little unclear and an illustration would certainly help. Does a “stationary contraction” refer to a single activation burst? Line 229: this sentence is very difficult to understand. During forward movement, when was the muscle active (stance phase and/or swing phase)? Was it a single burst throughout the stride or was a biphasic burst apparent? During destabilisation, were both left/right muscles activated and if so, when did this occur during the exercise? Descriptions of muscle activity must be clarified, especially if they are not illustrated in a figure.

Line 231 – how was “nonsense” amplitude defined? Were these not attenuated during post-processing or during the quality control checks of signals that are described early in section 2.5? What was done to account for these nonsense signals within the dataset? Were they removed as outliers or left in the analysis? How does this impact the results?

Figures 3 and 4: are these activation maps from individual/representative horses/trials? If so, how were these horses/trials chosen? It’s unclear to me how these data were analysed on a group level. Were they, or is this study more of a single case design? This should be made clearer in the manuscript, as the results section raises a lot of questions for me about this. What does the time domain on the x axis include for both figures? For Figure 3,  does the x axis to include a stance and swing phase of the left limb? If so, can the authors illustrate these phases to show how the muscle activates in relation to the actual task or at least include this information in the figure description? The same question goes for Figure 4, does the time domain relate to the initiation of lateral flexion --> peak lateral flexion --> and then a return to the starting position (i.e. the full ROM)? This needs to be illustrated and/or described. Figure 3 includes 4 subpanels, but 2 are labelled as “(a)”, is this an error?

Figure 5.  includes 3 subpanels, but 2 are labelled as “(a)”, is this an error? What do “n.1”, “n.2”, “n.3” refer to in each sub panel? Are these different horses/trials or one representative horse and different trials for mastication? This is unclear and needs to be specified? What does a chewing cycle refer to? One chewing cycle or multiple? Surely 20 seconds makes up more than one chewing cycle, so this must be clarified.

Line 272: what do the authors mean by “clip on the skin”?

Line 287: Again, I struggle to believe that the dynamic (although largely stationary) tasks performed in this study could elicit a fatigue response in these horses. There are also issues with measuring fatigue using the power spectra of an sEMG signal from non-isometric contractions. However, I can see that the authors address this briefly in the discussion section. The authors should provide information about the time required to conduct data collection and the number of repetitions for each task, which will help the reader to critically evaluate/interpret the described fatigue response.

Figure 6. again, it is unclear what the time domain on the x-axis refers to? Is it the full ROM for the lateral flexion exercise? Is subpanel c labelled incorrectly as RMS? This figure again raises questions about how the signals were filtered (RMS window length?), as this is not described in the methods section, as I’ve stated above. Is this one representative horse/trial? How/why was it chosen? Figure 7 raises similar questions to those that I’ve described for each of the above figures.

Discussion:

In my opinion, the discussion section is the strongest section in the manuscript. It reiterates that the aim of the study is to assess the feasibility of HD-sEMG in horses, but this clarity needs to be included in the introduction too. Grammar and sentence structure should be reviewed to improve readability throughout. I really feel like some of the discussion is better suited to the introduction section, as this section provides much more clarity about the study aim and design than the introduction does (i.e. parts of paragraph 1 of the discussion would be quite useful in the introduction). I don’t feel that I can provide more feedback on the discussion section until my above concerns are addressed.

Line 338: power or amplitude for RMS?

Line 395: human not “humane”

Author Response

Rebuttal Letter_Animals-1582664

Reviewer 1

This is an interesting study that evaluates the feasibility of HD-sEMG for objectively monitoring/measuring muscle activation in equine subjects, with a specific focus for its use in equine rehabilitation. Indeed, this study is required as a foundation for evaluating the usefulness of this technology for horses and I would really like to see this work published. However, in it's current form, I felt that the manuscript lacks a clear focus, flow, and structure. As an experienced reader/researcher in this field, it is clear to me that the author’s main aim is to evaluate the feasibility of HD-sEMG for use in horses, but I do not feel that this will always be clear to the readership. A number of variables are employed to fulfil the aims and these are calculated from many tasks, some of which seem very unrelated (i.e. mastication and dynamic mobilisation exercises used for equine rehabilitation/evaluation). The manuscript is missing quite a lot of information which is required to ensure its reproducibility and reading through it generally raised more questions than answers for me (some of which were answered much later in the manuscript). As such, I feel that this manuscript requires major revisions, where the authors can focus on presenting the reader with a clear aim, a clear justification for hypotheses and methods and a methodology section that ensures full transparency for reproducibility. Please see below an extensive list of my concerns/feedback for improvement. I hope that these are helpful and can ensure that this work is published. 

Author’s response_ Thank you to the reviewer for the positive comments and the extensive and accurate work performed on the manuscript. We are aware that this attentive review gives us the opportunity to improve our manuscript. The abstract and the introduction of the study were rewritten to improve background description and to improve the focus and the flow of information given in those sections. The figures describing the motor tasks and were reformatted to add information which are useful to reproduce the work using this technology. Some paragraphs (the 2.2 and the 2.5) were changed, to give a technical detail useful to reproduce our results. We are aware that this is a new technology a bit “unfriendly” at the beginning, due to the large number of information given in a short time. We hope in the current form the manuscript is more understandable.

Introduction: Sentence structure and grammar should be reviewed throughout the introduction section. I found some sentences were difficult to follow due to grammatical errors, missing words, or run-on sentences. For example: the explanation of HD-sEMG.

Author’s response_ A grammar and language review has been perfomed.

The flow of the introduction section could be improved, specifically the author’s presentation of the background/gaps in knowledge for the implementation of HD-sEMG, how this differs from bipolar EMG (commonly used in equine studies), the lack of standardization in the field of equine sEMG (the authors have missed some references here from St. George et al., who describe the lack of standardisation and have conducted research on this), applications for sEMG within equine rehabilitation. As it stands, each of these issues are raised to the reader, but there is a lack of flow, which links the ideas together and thus it may be difficult for the reader to understand how this impacts the current study’s aims and objectives/why this study is necessary. The authors could consider re-arranging some of the paragraphs so that the readability of the section is improved. My suggestions for this would be to first introduce the gaps in knowledge for objective monitoring/measurement techniques in equine rehabilitation (lines 77 – 87), then introduce the reader to how sEMG technology can be employed to fulfil the need for objective measures, followed by a review of current literature that has employed sEMG in horses, highlighting the lack of research that has employed HD-sEMG in horses (and how this differs from sEMG) and our current gaps in knowledge for standardising equine sEMG acquisition, processing and analysis. This can then lead to the aims and objectives of the study and how these will be achieved. I think that this would greatly improve the section.

Author’s response_ The introduction’s section was rewritten. The need for the introduction of objective monitoring techniques in equine rehabilitation is mentioned. The gap of knowledge for the implementation of HD-sEMG is now described in comparison to the current body of literature on bipolar sEMG in horses. The previously missed references from St.George et al. were added to the manuscript. Introduction section was reformatted following the suggestion of this review.    

Lines 93 – 100: This paragraph requires some attention and improvement. The aim is vague and does not describe the selected motor tasks or mention the dynamic mobilisation exercises (line 96 – 97) that will be used to fulfil it. There are a multitude of dynamic mobilisation exercises, which target different areas/movements in horses, but these are not described in detail anywhere in the introduction (apart from a brief mention in lines 82 – 85). Further, a reference referring to their description is not provided to the reader, apart from [9], which is a book and the reader cannot be expected to read an entire book to access this information. I would encourage the authors to provide more detail on the specific exercises/muscles that they are focussing on in this study and how they are performed, earlier in the introduction section. Further, are the authors not trying to evaluate the feasibility of this technology for use in horses/equine rehabilitation? If so, then this would add clarity to the study aim. The hypotheses are not clearly related to what has been described earlier in the introduction section and the aim of the study. With respect, the hypotheses seem to come up at random and I feel that this could be mitigated by ensuring that the reader is provided with more information within the introduction section that relates to each hypothesis (i.e. ability of HD-sEMG to measure muscle fatigue and dynamic mobilisation exercises in horses). 

Author’s response_ An extensive description of the dynamic mobilization exercises used in equine rehabilitation was included in the introduction. The hypotheses of the study are better described, giving to the reader not familiar with electromyographic technology more information to understand the reasons of this study.

Materials and Methods:

Line 103-106: the authors should provide a description of the metrics for the sample (i.e. horse age, breed, sex, height, weight, etc). Further information on each horse's fitness and work level is required. These are all important information in relation to the type of data collected and to ensure repeatability of the work.

Author’s response_ The information required by the reviewer are now added in a Table, at the beginning of the results section.

Section 2.3: Here I can appreciate that the authors have now provided further information about the dynamic mobilisation tasks that were performed in the study, but I would prefer it if (at least) some of this information were to appear earlier in the manuscript, as well as in this section. This would mitigate the questions that arise for the reader and make them aware of the methods that will be used to fulfil the aim of the study. It’s still unclear to me how mastication is used as a control task within the study. What do these muscles/chewing task control against?

Author’s response_ Description of dynamic mobilization in horse are now extensively described in the introduction, as suggested. Concerning the masseter muscles, authors are aware that mastication is a different motor task than motor task performed during dynamic mobilization exercises. Masseter muscle was instrumented because its activation is certainly obtained during mastication, and masseter is a superficial muscle easy to be instrumented with HD-sEMG grids. The activation of this muscle is simply to be realized by feeding a horse with good quality hay or a carrot. The spectrum of activity of the masseter muscle in this study is useful to demonstrate the effectiveness of the HD-sEMG system. This is a first validation study, therefore one of the aim of the study is to show that we are able to capture skeletal muscle activation. This is the reason because this muscle was considered in this study as a “positive control”. This is further explained in the discussion section.

The paragraph consisting of lines 132 – 135 could be removed or the authors could consider informing the reader that each of the tasks are described in detail in the proceeding paragraphs. Please include additional references for the justification for muscles/tasks studied in lines 126 – 131 and throughout this section. To ensure reproducibility could you provide further information for whether EAO activation tasks were elicited using a reward (i.e. carrot stretches)? SDP electrode grid location is missing in the text and not included in Figure 1.  Are SDP forward/backward tasks just one step in walk? Or consistent forward steps and backward steps? How many repetitions were required/collected for each horse/task?

Author’s response_ An extensive description of motor tasks employed in this study was added in the manuscript at the introduction section. Also, the method to obtain collaboration of the horses were better highlighted. The figures 2, 3 and 4 illustrate now the location of the bidimensional grids to capture the sEMG signals in the different muscles (SDP, EOA, masseter). Also, an image of the movement of the horse to activate those muscles were added (video were too large in term of megabyte to be introduced in the figures). This information was missing in the previous version of the manuscript. Details of the motor tasks are now better described in the captions of the figures. Concerning the SDP activation: the figure 3 describe the activation of the muscle during a step forward, the position of the grid was visualized as an additional panel in the figure. An additional video was added to show the SDP activation during the lateral destabilization exercise (as an additional material). Concerning the repetitions of the motor task, all the exercises proposed in this manuscript are prolonged muscle contraction, meaning that the exercise is sustained for many seconds. This is better explained in the figure captions and in the manuscript.  

Lines 172 – 174: This sentence is interesting. If these tasks have not yet been recorded using sEMG/HD-sEMG how did the authors define “nonsense” amplitudes or patterns? Surely, high-pass filtering protocols could be employed to attenuate low-frequency noise sources from movement artefacts? How many attempts/task repetitions were required for each horse to obtain quality signals? More information required here. 

Author’s response_ Nonsense amplitudes channels are defined by visually observe “mute” electromyographic signals during acquisition. If a mute channel is observed, grid was replaced, and task repeated until good quality signal was obtained. Collected data were post-processed with a band-pass digital filter (20 - 400 Hz) to eliminate high frequency noise and low frequency fluctuation due to artifact movements. The number of attempts for each horse to obtain quality signals have not been registered. If the electromyographic channels are working correctly (no mute channels) acquisition do not require large number of repetitions of the motor task. Concerning the effectiveness of the high-pass filter protocols employed: observing the map during the muscle contraction is an effective method to be sure that the signals is a muscle contraction vs an artefact. Muscle contraction appears like a wave of contraction starting in the region of the maps and migrating peripherally, with the activation moving like “the waves that spread from a stone thrown into a pond”. Artifacts appear like “patchy activation under a map” or a “point of activation that does not migrate”.

Section 2.5: This section raises a lot of questions for me and areas where I think further information should be provided to ensure reproducibility of the work. Currently there is not enough information for me to reproduce the post-processing and analysis: These include:

  • Line 180: how were “optimal signals” defined? What was the “defined study period”?
  • Line 181: I would argue that this band-pass filter would attenuate more than just environmental interferences. Movement artefact, cable motion artefact, etc?
  • Lines 184 – 187: the authors could consider referring the reader to some of the existing figures here, to illustrate the different methods/modes for analysis.
  • Line 187 and 195: milli volts or micro volts? Micro volts presented throughout the remainder of manuscript.
  • Line 188: please describe the “further filtering” that was conducted to attenuate noise sources. 
  • 189: Further information required here. Was the method described in [20] employed? This is not clear.
  • Line 191: “and based on an algorithm, described in [21]”?
  • Line 192: “can decompose at least 20 seconds”
  • Line 195: does this paragraph refer to the single differential signals? What was the time domain for each of these variables (RMS, MNF, MDF), so how were the signals analysed in relation to time/the dynamic mobilisation tasks? How can RMS, MNF and MDF be applied to detect the onset of muscle fatigue (shift in median frequency?). Further explanation, with appropriate referencing required here.
  • Line 202: again, here the author’s raise questions that are then addressed in later paragraphs (here they address my questions above about the onset of muscle fatigue). To correct this, combine paragraphs 195 – 206 and explain cohesively.
  • Line 204: what reference values were used for normalisation? The reader should not have to read an entire referenced work to obtain information that can easily be included in the text. Please include this information. 
  • Line 207 – 208: remove this sentence. Again, it just raises questions about whether the authors actually used correlation analysis, which is then answered in the paragraph below (line 209 – 210, which should be combined into one paragraph).
  • Can the authors realistically expect the tasks that are performed to induce fatigue? If not, why study this?

Author’s response_ The 2.5 section was completely rewritten to answer the questions raised by the reviewer on this section. Concerning the muscular fatigue, motor tasks used in this study are “sustained contraction”, meaning that horse were forced to maintain the position of neck and trunk flexion along 20 seconds using a reward (carrot). Therefore, I’m not surprise that sEMG data shows a muscular fatigue, because sustained prolonged muscle contraction along many seconds is a demanding exercise.

Results:

Line 220 – 221: the authors do not provide any explanation within the methods section about how onset, offset and activity duration were to be assessed. It appears that this was just visually/subjectively assessed, and if so, this should be clearly stated.

Author’s response_ Thanks to the reviewer for the inquiry. Using HD-sEMG the onset and duration of the muscle activity could be visually assessed during the exercise thank to the opportunity to observe the electromyographic map of the instrumented muscle. A remote registration of the activity is also possible when the signals are registered during the activity, for example if the acquisition are performed during the locomotion. Now, to reduce bias related to sweating and the loss of grid adhesion to the skin, we evaluated HD-sEMG during motor tasks that could be realized in a horse standing still. Signals acquired by the electrodes of the grid are correlated to the muscle activation of the muscle fibers under the skin, therefore the duration of the muscle activity is not “subjectively” only assessed, but it is an objective parameter. The goodness of the acquisition is subjectively and visually assessed to exclude artifacts in the acquisitions of electromyographic signals.

Line 222: “Masseter muscles raw signal present burst-like muscular activity during the chewing cycles” – this does not make sense to me. I do not know what the authors are trying to describe. What is the chewing cycle and how many bursts were observed within this cycle? More information required, as it is not a clear description of muscle activation. Figure 2 is very small and thus it is difficult to interpret such a large amount of information. How many chewing cycles are shown here over the approx. 30 second period? How were the epochs of muscular activation differentiated from the other clear (albeit shorter/lower amplitude) bursts of activation that occur throughout the cycle? Are these data from one representative horse? Further information required in the figure description.

Author’s response_We improperly used the term chewing cycle. This was corrected. The electromyographic signals were recorded during the mastication of a carrot, from the moment when the horse taken a carrot since the horse swallow the bolus, this is the reason of a recording time long 30 seconds. During this time the horse activate masseter muscle many times (20/23 times) producing a signal of this activation in the map. The epoch of the muscle activation selected to observe a map is 0.5 sec in our study. In the figure only one horse is represented. A small number of frames (7/8 frame representing each a single millisecond) are shown for each motor task and muscle activation. The selection of the epoch of the map is performed on the entire (single or double differential) electromyographic traces recorded with the grid. To observe the activation, we need to slow-down the video from real time to appreciate the pattern of muscle activation. This is better shown in the additional video S1.

Lines 224 – 233: The authors could consider providing figures for these tasks/exercises, as was done for mastication in Figure 2. Again, the explanations of muscle activation remain a little unclear and an illustration would certainly help. Does a “stationary contraction” refer to a single activation burst? Line 229: this sentence is very difficult to understand. During forward movement, when was the muscle active (stance phase and/or swing phase)? Was it a single burst throughout the stride or was a biphasic burst apparent? During destabilisation, were both left/right muscles activated and if so, when did this occur during the exercise? Descriptions of muscle activity must be clarified, especially if they are not illustrated in a figure.

Author’s response_Figures are now reformatted to improve clarity and relationship between the motor task and the muscular activation visualized in the maps. The motor tasks performed are different in term of mechanism for the different muscles. An isotonic contraction at the beginning of the task and a isometric contraction during the rest of the task characterized the lateral flexion of the trunk. This mean that the activity of the abdominal muscles during the head/neck flexion is a prolonged exercise characterized by an initial isotonic and eccentric contraction (head and neck flexion to the hip), followed by a isometric contraction (sustained flexion). The time domain of this exercise relates to the initiation of lateral flexion to the peak lateral flexion, and it does not include the return to the starting position. During destabilization exercise muscle contraction was essentially isometric for the muscle of the thoracic girdle against the pressure applied by the operator on the wither of the horse. This is the same during forward and backward exercises. Only a single step was registered during the stance phase of the stride. “Stationary contraction” is an improper term meaning a prolonged sustained isometric contraction.

Line 231 – how was “nonsense” amplitude defined? Were these not attenuated during post-processing or during the quality control checks of signals that are described early in section 2.5? What was done to account for these nonsense signals within the dataset? Were they removed as outliers or left in the analysis? How does this impact the results?

Author’s response_The non-sense amplitude channel are mute channels. The reason is ever a loss of contact between the skin and the grid. When a non-sense channel was present we did not “simply” remove the signal because this impact on the double differential electromyographic signals of the entire recording session. Therefore, we performed another acquisition after a new installation of the grid.

Figures 3 and 4: are these activation maps from individual/representative horses/trials? If so, how were these horses/trials chosen? It’s unclear to me how these data were analysed on a group level. Were they, or is this study more of a single case design? This should be made clearer in the manuscript, as the results section raises a lot of questions for me about this. What does the time domain on the x axis include for both figures? For Figure 3,  does the x axis to include a stance and swing phase of the left limb? If so, can the authors illustrate these phases to show how the muscle activates in relation to the actual task or at least include this information in the figure description? The same question goes for Figure 4, does the time domain relate to the initiation of lateral flexion --> peak lateral flexion --> and then a return to the starting position (i.e. the full ROM)? This needs to be illustrated and/or described. Figure 3 includes 4 subpanels, but 2 are labelled as “(a)”, is this an error?

Author’s response_ As previously said, figures are now reformatted to improve clarity of the acquisition and not only the maps of muscle activation are shown. Authors hope that in the current form figures are more understandable to the readers.

Figure 5.  includes 3 subpanels, but 2 are labelled as “(a)”, is this an error? What do “n.1”, “n.2”, “n.3” refer to in each sub panel? Are these different horses/trials or one representative horse and different trials for mastication? This is unclear and needs to be specified? What does a chewing cycle refer to? One chewing cycle or multiple? Surely 20 seconds makes up more than one chewing cycle, so this must be clarified.

Author’s response_Sorry it was a typo error. Concerning the trial, it was three mastication in a single trial in a single horse

Line 272: what do the authors mean by “clip on the skin”?

Author’s response_ The wording has been changed to improve clarity. Now the text reports: “Skin preparation is essential for an optimal recording of low noise levels electromyography signals using multiple arrays electrode grids [16]. Skin preparation was performed to assure the complete removal of hairs and flaky skin layers to reduce the electrical impedance at the level of skin-electrodes interface and to obtain proper electrodes adhesion [17]. Skin was clipped, shaved, washed with a neutral soap solution, and further cleaned with an abrasive cream”

Line 287: Again, I struggle to believe that the dynamic (although largely stationary) tasks performed in this study could elicit a fatigue response in these horses. There are also issues with measuring fatigue using the power spectra of an sEMG signal from non-isometric contractions. However, I can see that the authors address this briefly in the discussion section. The authors should provide information about the time required to conduct data collection and the number of repetitions for each task, which will help the reader to critically evaluate/interpret the described fatigue response.

Author’s response_ As previously described, fatigue was observed after a prolonged activation rather than a repeated activation. In practice, the horse was asked to bend its neck and to keep the neck in the same position for a prolonged time during which sEMG signals were acquired. This also responds to the reviewer concern regarding the measurement of fatigue in non-isometric condition, as fatigue was measured during “almost-isometric” (compatibly with in vivo measuring) muscle activation/contraction. Concerning the time required to the muscle to reach fatigue, properly defined muscle fatigue cannot be reached if we considered fatigue as a reduction in force (drop in the generated newton). This is because we have not the opportunity to capture the force generated in newton during proposed motor tasks. Our fatigue evaluation is strictly related to the electromyographic definition of fatigue (Yousif et al. 2019 IOP Conf. Ser.: Mater. Sci. Eng. 705 012010; Assessment of muscle fatigue based on surface EMG signals using machine learning and statistical approach: A Review.)

Figure 6. again, it is unclear what the time domain on the x-axis refers to? Is it the full ROM for the lateral flexion exercise? Is subpanel c labelled incorrectly as RMS? This figure again raises questions about how the signals were filtered (RMS window length?), as this is not described in the methods section, as I’ve stated above. Is this one representative horse/trial? How/why was it chosen? Figure 7 raises similar questions to those that I’ve described for each of the above figures.

Author’s response_ The time domain on the x-axis refers to entire activation of the abdominal muscles during a lateral flexion, and it is related to the initiation of lateral flexion to the peak lateral flexion without the return to the starting position. The subpanel C is labelled incorrectly, this is the MDF not RMS. The signal was filtered using a high-band filter as described previously. The trial refers to one horse/one trial. This was chosen because it was a representation of our data. 

Discussion:

In my opinion, the discussion section is the strongest section in the manuscript. It reiterates that the aim of the study is to assess the feasibility of HD-sEMG in horses, but this clarity needs to be included in the introduction too. Grammar and sentence structure should be reviewed to improve readability throughout. I really feel like some of the discussion is better suited to the introduction section, as this section provides much more clarity about the study aim and design than the introduction does (i.e. parts of paragraph 1 of the discussion would be quite useful in the introduction). I don’t feel that I can provide more feedback on the discussion section until my above concerns are addressed.

Author’s response_Large part of the discussion section is now presented in the introduction as suggested.

Line 338: power or amplitude for RMS?

Author’s response_ The RMS amplitude format is more correct probably, because RMS indicates the equivalent steady state energy value of an oscillating signal.

Line 395: human not “humane”

Author’s response_Done

Reviewer 2 Report

Thank you for preparing this interesting and novel manuscript. I have few concerns of significance. To improve the manuscript i would suggest

1) to expand the discussion of how the technology described can reflect muscle fatigue - this is not clear to someone not extremely knowledgeable of the technology

2) to show (our point out somehow) in similar graphics absence of fatigue in the masseter muscle ( which i believe was not seen?)

It is unclear to this reader also how this technique demonstrates a reduction in force of muscle contraction -- expanded discussion to highlight this would be a useful improvement. 

Author Response

Reviewer 2

Thank you for preparing this interesting and novel manuscript. I have few concerns of significance. To improve the manuscript i would suggest

1) to expand the discussion of how the technology described can reflect muscle fatigue - this is not clear to someone not extremely knowledgeable of the technology

2) to show (our point out somehow) in similar graphics absence of fatigue in the masseter muscle ( which i believe was not seen?)

It is unclear to this reader also how this technique demonstrates a reduction in force of muscle contraction -- expanded discussion to highlight this would be a useful improvement. 

Author’s response_ Thank you to the reviewer for these inquiries about muscle force and fatigue. This is a very interesting field still open to many questions. I think the discussion on determination of muscular fatigue based on electromyographic signal is very interesting and still unresolved. The discussion session was reformatted and the problem better explained.

The single most relevant determination of fatigue is done through the measurement of force or power measurement, which is produced during a voluntary effort of maximum intensity, meaning maximal voluntary contractions (MVCs) test. In general, when the subject performs the task of interest the drop of maximal force output from a particular muscle is the method to show fatigue. The force output decline rate measured in MVCs tests will indicate the fatigue. Similar tests which assist in the direct evaluation of force are not possible in animals. Using EMG, the changes of sEMG characteristic values, such as decrease of the median power frequency (MDF) and increase of the root-mean-square (RMS), are very often used to estimate fatigue. But it should be noted that these sEMG indicators are not always applicable. As De Luca puts it, the usage of spectral variables should be limited to isometric, constant load tasks greater than 30% of subjects’ maximal force. In addition, the sEMG amplitude is significantly affected by muscle force: greater force leads to larger amplitude. It seems the RMS should only be used in constant force fatigue process.

In our study the analysis of sEMG spectral features corroborate the research in human field, where the total RMS increases during a isometric contraction (like the sustained lateral flexion of the head and neck in our study) and the MDF decrease. We are not able to produce a similar constant-force fatiguing tasks for the masseter muscle. The activation of this muscle during mastication is a series of burst activity and not a isometric contraction. Therefore, I suspect that is not correct to show the sEMG spectrum of the masseter over many mastication as a representation of fatigue/not fatigue. This is a correct representation for the lateral flexion because it is an exercise at the beginning with a concentric and isotonic muscle contraction (when the horse start to flex the head and neck to the final position at the level of the hip), and it is followed by a isometric contraction (maintaining the position due to the present of a carrot). With a good approximation this task is comparable to an isometric contraction. But further study on proper motor tasks are essential to produce consistent results on this topic. This is clarify in the discussion, that more research are needed to demonstrate the effective application of this technology to the determination of muscle fatigue.

Reviewer 3 Report

Dear Authors,

The study evaluates the feasibility of HD-sEMG for analizing muscle activation in the horse. It is intended as a preliminary to fine-tune the technique and to have parameters for comparison. HD-sEMG is a tool that, once developed definitively, can provide usable data both in the clinic and in the research of the sport horse.

The topic is very technical and not easy to read but opens up important developments to better understand muscle fatigue. The research is new and original even if it needs further investigation.

The conclusions are sustainable even if they leave open questions that will have to be investigated in further research.

References are appropriate, although a few items could have been added.

In my opinion tables and figures are sufficiently described.

The authors could improve the description of muscle work measurement techniques to standardize the acquisition, processing and analysis of equine sEMGs.

In my opinion the experimental study is important and well conceived but, as stated in the conclusions, it needs further investigations. The results are interesting but to be assessed following further necessary findings.

Author Response

Reviewer 3

The study evaluates the feasibility of HD-sEMG for analizing muscle activation in the horse. It is intended as a preliminary to fine-tune the technique and to have parameters for comparison. HD-sEMG is a tool that, once developed definitively, can provide usable data both in the clinic and in the research of the sport horse.

The topic is very technical and not easy to read but opens up important developments to better understand muscle fatigue. The research is new and original even if it needs further investigation.

The conclusions are sustainable even if they leave open questions that will have to be investigated in further research.

References are appropriate, although a few items could have been added.

In my opinion tables and figures are sufficiently described.

The authors could improve the description of muscle work measurement techniques to standardize the acquisition, processing and analysis of equine sEMGs.

In my opinion the experimental study is important and well conceived but, as stated in the conclusions, it needs further investigations. The results are interesting but to be assessed following further necessary findings.

Author’s response_ Thank you to the reviewer for the comments. We are aware that this is a prelaminar work to demonstrate the application of this new promising technology in the field of equine rehabilitation. In human HD-sEMG is a well demonstrated technology with strong application in many fields. A further effort is required to obtain consistent data on muscle contraction in horses.

Actually, the manuscript had been extensively reviewed to make the text more fluent and to show how this technology could be reproduce in further research. The abstract and the introduction of the study were rewritten to improve background description and to improve the focus and the flow of information given in those sections. The figures describing the motor tasks were completely reformatted to add information which are useful to reproduce the work using this motor tasks. Paragraphs 2.2 and 2.5 were rewritten, to give a technical detail useful to reproduce our results. We are aware that this is a new technology a bit “unfriendly” at the beginning, due to the large number of information given in a short time. We hope the manuscript is more understandable in the current form.

Reviewer 4 Report

The paper seems not appropriate for Animals. The subject is very technical and should be submitted for example to Sensors. The paper is rather preliminary study and it should be underlined in the tittle. The paper is based on low amount of unknown horses, their characteristics is lacking. It is not known what are the limitations of  measurement (for example breed, skin coat).

There are many others unknown and unclear data like – technical details of all equipment (amplifier characteristics, logger description etc.). Information on signal transmission seems limited, possible signal lost not mention. The device should be better described, especially if it is the first work on it (?). It seems promising, but more information is required.

The other problem is not sufficient description of the horse motor trials. It is not given how many strides were performed, how long the exercise was performer. In L 104 it is given that three trials were done. The repeatability/comparability of data is not presented. Except figures the basic statistic of results should be given in table or in the text.

The R-square is low (below 0.3) in many cases, so many factors are not taken into account. The data from figure 3 are extremely low, that was not discussed.

The paper should be reconsidered preferably by other journal after major revision.

The detailed description of all parts is missing – horses not described, equipment not precise, exercise not characterised. That paper does not allow for its repetition because of missing methodology.

Results should be presented more detailed. Fatigue control should be investigated more detailed, Discussion is good, however the connection with literature weak. The paper is promising, however should be re-written.

The comparison with other methods would help for better presentation.

Author Response

The paper seems not appropriate for Animals. The subject is very technical and should be submitted for example to Sensors. The paper is rather preliminary study and it should be underlined in the tittle. The paper is based on low amount of unknown horses, their characteristics is lacking. It is not known what are the limitations of  measurement (for example breed, skin coat).

There are many others unknown and unclear data like – technical details of all equipment (amplifier characteristics, logger description etc.). Information on signal transmission seems limited, possible signal lost not mention. The device should be better described, especially if it is the first work on it (?). It seems promising, but more information is required.

The other problem is not sufficient description of the horse motor trials. It is not given how many strides were performed, how long the exercise was performer. In L 104 it is given that three trials were done. The repeatability/comparability of data is not presented. Except figures the basic statistic of results should be given in table or in the text.

The R-square is low (below 0.3) in many cases, so many factors are not taken into account. The data from figure 3 are extremely low, that was not discussed.

The paper should be reconsidered preferably by other journal after major revision.

The detailed description of all parts is missing – horses not described, equipment not precise, exercise not characterised. That paper does not allow for its repetition because of missing methodology.

Results should be presented more detailed. Fatigue control should be investigated more detailed, Discussion is good, however the connection with literature weak. The paper is promising, however should be re-written.

The comparison with other methods would help for better presentation.

Author’s response_ Thank you to the reviewer for the comments. We are aware that this is a prelaminar work to demonstrate the application of this new promising technology in the field of equine rehabilitation. In human HD-sEMG is a well demonstrated technology with strong application in many biomedical fields.

Authors prefer to show this manuscript in Animals to a veterinary audience than on Sensors, which contemplate an engineering-based audience. 

The paper was extensively revised. The abstract and the introduction of the study were rewritten to improve background description and to improve the focus and the flow of information given in those sections.

The figures describing the motor tasks were reformatted to add information which are useful to reproduce the work using this technology. An extensive description of the motor tasks used in horse was provide in the introduction to better explain the work to readers interested in rehabilitation.

Paragraphs 2.2 and 2.5 were rewritten, to give a technical detail useful to reproduce our results.

Concerning the low number of horses employed in the study, instrumentation of muscle was performed on a small sample of animals which now are better described in a demographic table.

Comparison of HD-sEMG with other methods to measure EMG is not feasible now, because it is impossible to compare simple bipolar sEMG with data derived from HD-sEMG in this sample of animals. Concerning the difference between the two techniques, in the introduction we have extensively describe this point now.

I hope the manuscript could be reconsidered after this revision.

Round 2

Reviewer 1 Report

The authors should be applauded for their extensive efforts to improve the manuscript. I would like to thank them and commend them on their efforts to address each of my previous comments. Although I feel that the manuscript is much improved, there is a clear lack of attention to detail in certain areas, and the manuscript requires some minor revisions to ensure that the manuscript is of high-quality. This is especially important for this manuscript, which represents the first study of its kind and it is therefore likely to become a key article. As such, it is imperative that the authors review the manuscript with a great attention to detail, paying particular attention to a revision of English language, grammar and sentence structure. I have provided below another list of suggestions, which I still feel need to be addressed. As stated previously,  this study is required as a foundation for evaluating the usefulness of this technology for horses and I would really like to see this work published.

The simple summary leaves me with a bit of a “so what?” feeling? The actual importance and originality of this study only comes in at line 25 and the lines before this, in my opinion, are of little value – mainly because the authors never describe the limitations of these existing tools and why further work on objective methods are required. Nothing is mentioned about the exercises used to test the technology in horses and how this can inform objective methods for evaluating or tracking physiotherapy treatments over time. This is an important part of the manuscript and it is important that the authors make the most of this opportunity to describe the impact of their manuscript

The abstract is improved, but mentions nothing about equine physiotherapy/rehabilitation applications? This is an important aspect throughout the manuscript and indeed the reader is exposed to this in the simple summary, so why not include it in the abstract?

Line: 38 – I much prefer the original sentence, which includes “evaluate the feasibility of HD-sEMG for measuring equine superficial skeletal muscle contraction during specific motor tasks”. The revised sentence is less clear.

Line 40: to fulfil this aim, activation of the superficial... were measured using HDsEMG during dynamic mobilization exercises to induce lateral bending and flexion/extension tasks of the trunk. Can a little more detail be added about the specific tasks mentioned in the sentence above?

Line 44: as a control condition.

Throughout the abstract and manuscript: consider the use of “contraction” vs “activation”. Is the muscle actually contracted throughout the entire task or is activation a better overarching term? Seems to me that you don’t only measure active contractions?

Line 50: permitting inferences about the motor control strategy.

I do think that the authors need to (ideally abstract, simple summary and intro) clarify that this is a preliminary study, so the majority of the analysis is descriptive/subjective in nature. I think the authors should consider specifying the preliminary nature of the study in the title too. “A preliminary study on the use of HDsEMG for the functional imaging of equine superficial muscle activation during dynamic mobilization exercises”. Just a suggestion, but to me this better reflects the aim, objective and methodology of the study and is not so “concrete” in nature, as clearly more work needs to be done to confirm the current title.

Introduction: much improved – some specific suggestions for further improvement/readability.

Line 78: consider combining these sentences for more impact: of major importance, so the development of new tools for the objective assessment of …. “. Please mention sEMG in line 79, otherwise it’s a bit of a mute sentence and does not lead well into the next paragraph where it’s described.

Does the second paragraph refer to horses or other species? Please clarify, I’m a bit confused at times with this.

Line 85: please provide a reference for this. Is this a depth that you refer to? If so, then this is surely dependent on the IED? Same goes for Line 87? Where are these values coming from? Does this paragraph refer to human subjects or equine? It is not clear to me.

Line 92: amount instead of number of information detected by sEMG sensors. Language must be reviewed throughout. 

Line 99: consider concluding this paragraph with a statement to say that, to date, no studies have employed HD sEMG on equine subjects, so the feasibility of this technology as an objective measure of equine muscle activation remains unknown. This leads well into the next paragraph.

Line 101 – 108: I’m struggling a little with the relevance of this paragraph, especially as there are no clear conclusions, and this leads to a seemingly unrelated paragraph on existing works? Can the authors tie this paragraph up better and leave the reader with either a solution to how they will get around it in this paper or at least state that work is currently being done to develop standardised methods for equine sEMG? Or remove the paragraph altogether, as I’m not really convinced that it adds much (and lines 117 – 119 seem to cover this idea anyways).

Line 110 – 111: I prefer the original sentence, as it is becoming less “rare” but indeed less frequently employed when compared to human studies. I also prefer the original sentence for 110, as the review did not only look at the field of veterinary medicine, it included broader animal biomechanics papers (as stated in the review title).

Line 114: why does referencing stop after chewing?

Line 116: three studies but reference only one?

Line 151: again I would specify somewhere that you’re referring to equine core musculature.

Line 158: The sustained “chin-to-flank” lateral bending, “chin-between-knees” rounding (flexion?), and balancing/destabilisation exercises are …. The authors should check that they consistently use the same terminology for these exercises throughout. For example, chin between knees and chin between carpus are both used. Sometimes lateral bending exercise is used alone, etc. Please be consistent for the readership. 

Line 169: destabilisation exercise, where a forelimb is raised/picked up and pressure is applied to the sternum or point of shoulder to shift the weight caudally (please use appropriate anatomical terminology throughout), causing ….

Line 173: review this sentence please, realise is not an appropriate term and it’s unclear if this is a different exercise/why it’s being described. So there are 3 destabilisation exercises? Sternum pressure, withers pressure and foreward/backwards motion? It's still unclear why the authors describe these specific exercises in the intro: we know they are common, but will they be measured in this study? Further clarity for how these exercises relate to the current study would improve readability. 

Line 184, surely it is better to specify this in the aim? Instead of using selected motor tasks, state the dynamic mobilisation exercises, as described above. Or does selected motor tasks also encompass mastication? 

Line 177: please split this sentence up for clarity. It is run-on and difficult to follow.

Line 186: measured core muscles, as this is all you have information on.

Line 188: prolonged muscle contraction during dynamic mobilisation exercises. Please be clear about what you’re explicitly measuring/evaluating. Where is muscle force measured?

Methods:

Line 193: Please refer to Table 1 here instead of results.

Line 201: channels?

Line 204 - 208: please review grammar throughout: “allows to amplify” à amplifies signals at XX ….

Figure in 213 does not contain a figure legend? Or are the figures in lines 213 – 221 combined and distorted by the tracked changes?

Line 230: again please check grammar/wording of sentences.

Line 237: easy to record?

Line 263: the authors should specify that they gave the horse hay or a carrot and recorded the chewing of this reward. How is this standardised if horses were given 2 different types of feed, which arguably require differing movements and time to chew?

Line 266 – 269: attention to grammar required.

Line 280: I am still fairly uncomfortable with the term “non-sense amplitude or pattern” – if sites were cleaned and adhesion is good then why would some recordings produce “non-sense”. I would urge the authors do define how they assessed signal quality in more detail or at least better define their inclusion criteria for high-quality signals.

Line 272: a conductive cream was spread over the conductive pads and then sensors were applied to the subject?

Section 2.5 is much improved – thank you!

Line 313: that reduces. I would also advise the authors to clarify that muscle activation was assessed subjectively though visualisation of the images.

Line 321 – run-on and hard to follow. Please review and revise sentence. Please can the authors review again their grammar, sentence structure and working throughout, I will refrain from making more specific recommendations past this point.

Line 327: millivolts used here but microvolts presented in all figures and results text? Why?

Line 340: for reproducibility, please specify the reference value used.

Results:

Line 364: why is offset removed here? Did the authors not calculate this? If so, shouldn’t it be removed throughout (it is still present in earlier sections)? But then how do you calculate activation duration if there’s no offset? Surely activation duration is just dependant on how prolonged the exercise is? How can you standardise this?

Line 376: I’m still really struggling to follow this description. What is “burst like muscular activity”? A single burst? Multiple bursts? Do these occur during ipsilateral swing phase during the forward motion? Where does the reported swing phase occur during the destabilisation exercise? Needs to be more clearly described. No activation bursts were detected during the backwards motion? Was only baseline activity detected?  The authors state that they detected onset and activation duration, but this is not included in the descriptions for any of the muscles?

Figures 2 – 4 are very small and contain a lot of detail, so difficult to decipher, particularly the sEMG traces. Hopefully these will be much larger in the manuscript or could the authors consider presenting only the highlighted sections and then presenting figures with full traces a supplemental file?

Greater attention to detail is required for figure legends:

Figure 3 does not reference the subpanel b. In figure 3, based on the sEMG traces, where does the exercise begin and end? Noise is apparent in some signals especially on the right side? Was this particular epoch chosen specifically to avoid this? Figure 4 description must be reviewed – it has quite a few errors for example the reference to c is an image and this description is repeated incorrectly for colour maps (line 432).

Figure 5 – again, please review the description for clarity. What does three mastication refer to? Trials? Cycles? N 1 – 3 labelling also unclear. I understand that this is one representative horse, but this should be clarified for the reader.

Line 459: across all motor tasks? Including mastication or just the dynamic mobilization exercises studied? Please clarify.

Figure 7: one representative horse? The description for figure 8 should include the same detail as figure 7, but it’s a little unclear as to what has been deleted in terms of this figure, as the tracked changes appear to distort this area of the document.

Line 490: across all horses?

Can the authors consider including some tabular data for some of their variables? At the moment, the results presented are from individual subjects and are thus highly descriptive and subjective in nature. The ability to present some data from the entire dataset would greatly improve the work and conclusions drawn.

The discussion requires an extensive review of grammar, English language and sentence structure.

Author Response

The authors should be applauded for their extensive efforts to improve the manuscript. I would like to thank them and commend them on their efforts to address each of my previous comments. Although I feel that the manuscript is much improved, there is a clear lack of attention to detail in certain areas, and the manuscript requires some minor revisions to ensure that the manuscript is of high-quality. This is especially important for this manuscript, which represents the first study of its kind and it is therefore likely to become a key article. As such, it is imperative that the authors review the manuscript with a great attention to detail, paying particular attention to a revision of English language, grammar and sentence structure. I have provided below another list of suggestions, which I still feel need to be addressed. As stated previously, this study is required as a foundation for evaluating the usefulness of this technology for horses and I would really like to see this work published.

Author response: Thank you for your suggestions to improve the manuscript.

            The simple summary leaves me with a bit of a “so what?” feeling? The actual importance and originality of this study only comes in at line 25 and the lines before this, in my opinion, are of little value – mainly because the authors never describe the limitations of these existing tools and why further work on objective methods are required. Nothing is mentioned about the exercises used to test the technology in horses and how this can inform objective methods for evaluating or tracking physiotherapy treatments over time. This is an important part of the manuscript, and it is important that the authors make the most of this opportunity to describe the impact of their manuscript

Author response: The simple summary was rewritten, and the focus of the manuscript was better explained.

 An evidence-based approaches to assess the effect of rehabilitation protocols is currently under investigation in animal physiotherapy. Research aimed to identify objective outcome measurements are of major importance in this field. Functional muscle assessment remains challenging during the application of specific exercises to activate core muscles in horses. This study introduces for the first time the use of bi-dimensional High-Density Surface Electromyography (HD-sEMG) in horse as a non-invasive tool that provides a functional imaging of superficial skeletal muscle activation, and contraction. Using a 32 and/or 64 surface EMG electrode grids positioned on two superficial core muscles, a colour scale image obtained from recorded signals allowed visual assessment of muscle contraction. The exercises used to test this technology were the destabilization exercise and the chin to lateral bending exercise. Our results support the use of HD-sEMG in equine physiotherapy to obtain an immediate biofeedback of muscle activation and objectively tracking the physiotherapy treatments.”

            The abstract is improved, but mentions nothing about equine physiotherapy/rehabilitation applications? This is an important aspect throughout the manuscript and indeed the reader is exposed to this in the simple summary, so why not include it in the abstract?

Author response: The concept of rehabilitation exercise was now introduced in the abstract.

The goal of the study was to evaluate the feasibility of HD-sEMG to detect electrical activation of skeletal muscle and its application during rehabilitation exercises in horse. To fulfil this aim, activation of the superficial descending and external abdominal oblique core muscles were measured using HD-sEMG technology during dynamic mobilization exercises to induce lateral bending and flexion/extension tasks of the trunk.”

            Line: 38 – I much prefer the original sentence, which includes “evaluate the feasibility of HD-sEMG for measuring equine superficial skeletal muscle contraction during specific motor tasks”. The revised sentence is less clear.

Author response: The original sentence was introduced. 

            Line 40: to fulfil this aim, activation of the superficial... were measured using HDsEMG during dynamic mobilization exercises to induce lateral bending and flexion/extension tasks of the trunk. Can a little more detail be added about the specific tasks mentioned in the sentence above?

Author response: The sentence was modified accordingly to the reviewer’s suggestion.

            Line 44: as a control condition.

Author response: Done.

            Throughout the abstract and manuscript: consider the use of “contraction” vs “activation”. Is the muscle actually contracted throughout the entire task or is activation a better overarching term? Seems to me that you don’t only measure active contractions?

Author response: From an electromyographic point of view the correct term is “activation”. This term has been utilized throughout the abstract and the entire manuscript.

            Line 50: permitting inferences about the motor control strategy.

Author response: Done.

            I do think that the authors need to (ideally abstract, simple summary and intro) clarify that this is a preliminary study, so the majority of the analysis is descriptive/subjective in nature. I think the authors should consider specifying the preliminary nature of the study in the title too. “A preliminary study on the use of HDsEMG for the functional imaging of equine superficial muscle activation during dynamic mobilization exercises”. Just a suggestion, but to me this better reflects the aim, objective and methodology of the study and is not so “concrete” in nature, as clearly more work needs to be done to confirm the current title.

Author response: Thank you, we accepted the proposed title.

Introduction: much improved – some specific suggestions for further improvement/readability.

            Line 78: consider combining these sentences for more impact: of major importance, so the development of new tools for the objective assessment of …. “. Please mention sEMG in line 79, otherwise it’s a bit of a mute sentence and does not lead well into the next paragraph where it’s described.

Author response: Done

            Does the second paragraph refer to horses or other species? Please clarify, I’m a bit confused at times with this.

Author response: We better specified when we are referring to humans vs animals studies.

            Line 85: please provide a reference for this. Is this a depth that you refer to? If so, then this is surely dependent on the IED? Same goes for Line 87? Where are these values coming from? Does this paragraph refer to human subjects or equine? It is not clear to me.

Author response: We are referring to humans’ studies. The requested references are now provided in the text.

            Line 92: amount instead of number of information detected by sEMG sensors. Language must be reviewed throughout. 

Author response: Done

            Line 99: consider concluding this paragraph with a statement to say that, to date, no studies have employed HD sEMG on equine subjects, so the feasibility of this technology as an objective measure of equine muscle activation remains unknown. This leads well into the next paragraph.

Author response: Thank for the suggestion. Done.

            Line 101 – 108: I’m struggling a little with the relevance of this paragraph, especially as there are no clear conclusions, and this leads to a seemingly unrelated paragraph on existing works? Can the authors tie this paragraph up better and leave the reader with either a solution to how they will get around it in this paper or at least state that work is currently being done to develop standardised methods for equine sEMG? Or remove the paragraph altogether, as I’m not really convinced that it adds much (and lines 117 – 119 seem to cover this idea anyways).

Author response: Thank for the suggestion. The paragraph was removed.

            Line 110 – 111: I prefer the original sentence, as it is becoming less “rare” but indeed less frequently employed when compared to human studies. I also prefer the original sentence for 110, as the review did not only look at the field of veterinary medicine, it included broader animal biomechanics papers (as stated in the review title).

Author response: The original sentence was reintegrated.

            Line 114: why does referencing stop after chewing?

Author response: It was a typo error.

            Line 116: three studies but reference only one?

Author response: The correct reference was added, this is the review by Valentin et al. (J Electromyogr Kinesiol 2016)

            Line 151: again I would specify somewhere that you’re referring to equine core musculature.

Author response: Done

            Line 158: The sustained “chin-to-flank” lateral bending, “chin-between-knees” rounding (flexion?), and balancing/destabilisation exercises are …. The authors should check that they consistently use the same terminology for these exercises throughout. For example, chin between knees and chin between carpus are both used. Sometimes lateral bending exercise is used alone, etc. Please be consistent for the readership. 

Author response: We modified the text accordingly.

            Line 169: destabilization exercise, where a forelimb is raised/picked up and pressure is applied to the sternum or point of shoulder to shift the weight caudally (please use appropriate anatomical terminology throughout)

Author response: We modified the text accordingly.

            Line 173: review this sentence please, realise is not an appropriate term and it’s unclear if this is a different exercise/why it’s being described. So there are 3 destabilization exercises? Sternum pressure, withers pressure and foreward/backwards motion? It's still unclear why the authors describe these specific exercises in the intro: we know they are common, but will they be measured in this study? Further clarity for how these exercises relate to the current study would improve readability. 

Author response: Thank you for the observations. Destabilization exercises used in this study are the withers pressure an the foreward/backward motion. This is now better explained in the text.

            Line 184, surely it is better to specify this in the aim? Instead of using selected motor tasks, state the dynamic mobilisation exercises, as described above. Or does selected motor tasks also encompass mastication? 

Author response: The overlong description of the aim of the study was shortened and modified as suggested. Motor task do not encompass mastication, this activity was elicited to obtained data for a muscle which is activated offering food.

            Line 177: please split this sentence up for clarity. It is run-on and difficult to follow.

Author response: The sentence was revised.

            Line 186: measured core muscles, as this is all you have information on.

Author response: The sentence was revised.

            Line 188: prolonged muscle contraction during dynamic mobilisation exercises. Please be clear about what you’re explicitly measuring/evaluating. Where is muscle force measured?

Author response: The sentence was revised. A direct measurement of the muscle force is not feasible using HD-sEMG without the ability to produce a maximal voluntary contraction of the instrumented muscle. The measurement of the muscle fatigue is the consequence of the spectral analysis of HD-sEMG signals.

Methods: 

            Line 193: Please refer to Table 1 here instead of results.

Author response: Done

            Line 201: channels?

Author response: deleted and substituted with signal

            Line 204 - 208: please review grammar throughout: “allows to amplify” à amplifies signals at XX ….

Author response: sentence reformatted.

Amplified signals at bandwidth 10–500 Hz were registered and converted in digital values….”

            Figure in 213 does not contain a figure legend? Or are the figures in lines 213 – 221 combined and distorted by the tracked changes?

Author response: The figure 1 is distorted by the tracking. Probably is better to read the clean version of the manuscript.

            Line 230: again please check grammar/wording of sentences.

Author response: sentence reformatted.

Instrumented muscles were superficially positioned with limited superimposition with other muscle bellies.”

            Line 237: easy to record?

Author response: Done

            Line 263: the authors should specify that they gave the horse hay or a carrot and recorded the chewing of this reward. How is this standardised if horses were given 2 different types of feed, which arguably require differing movements and time to chew?

Author response: Horses received hay for mastication task, not carrot. Sorry for the misunderstanding.

            Line 266 – 269: attention to grammar required.

Author response: Done

            Line 280: I am still fairly uncomfortable with the term “non-sense amplitude or pattern” – if sites were cleaned and adhesion is good then why would some recordings produce “non-sense”. I would urge the authors do define how they assessed signal quality in more detail or at least better define their inclusion criteria for high-quality signals.

Author response: Thank for the request of clarification. Two artifacts are common with HD-sEMG requiring attentive evaluation of the signals: mute channel and non-sense amplitude recording, due to motion artifact (mainly movement of the skin due to cutaneous trunci muscle). The term non-sense was eliminated because it is unclear and better explained in the following sentence:

Each motor task was repeated until a clean electromyography signal was recorded without mute channels recording or skin-to-electrodes motion artifacts. This mean that a coherent appearance of the myographic signals need to be detected during motor tasks recording.”

            Line 272: a conductive cream was spread over the conductive pads and then sensors were applied to the subject?

Author response: Yes, the foam pads were spread on the free side with a conductive cream. This was explained in the text.

Section 2.5 is much improved – thank you!

            Line 313: that reduces. I would also advise the authors to clarify that muscle activation was assessed subjectively though visualisation of the images.

Author response: Correction is done. A sentence clarifying the subjectively nature of the image capture was introduced in the text:

Out of several trials, the better signals were recorded and archived based on subjective assessment of the operator though direct visualisation of the images.”

            Line 321 – run-on and hard to follow. Please review and revise sentence. Please can the authors review again their grammar, sentence structure and working throughout, I will refrain from making more specific recommendations past this point.

Author response: The sentence was reformatted and the less than clear statement at the end of the sentence eliminated.

            Line 327: millivolts used here but microvolts presented in all figures and results text? Why?

Author response: Sorry, microvolts is the right unit. This was used now all over the manuscript.

            Line 340: for reproducibility, please specify the reference value used.

Author response: This sentence on the myoelectrical pattern of muscle fatigue was difficult to follow, we rewrote the paragraph in a way that is more understandable to the reader.

“Myoelectrical signs of fatigue are identified by an increase in RMS values within time and a decrease in MNF and MDF values. This fatigue indicators pattern is related to the reduction of muscle fiber conduction velocity, which decreases more or less rapidly depending on the level of contraction and causes a change in the power spectrum of the sEMG toward the lower frequencies, with a decrement of the mean and median spectral frequencies [2, 26].”

Results: 

            Line 364: why is offset removed here? Did the authors not calculate this? If so, shouldn’t it be removed throughout (it is still present in earlier sections)? But then how do you calculate activation duration if there’s no offset? Surely activation duration is just dependant on how prolonged the exercise is? How can you standardise this?

Author response: Sorry, the removal of the term offset was an error during the previous revision. The term was now reintegrated in the text. Concerning the question if the activation is dependant on how prolonge the exercise is, we analyzed the sEMG signals during a motor task. The sEMG signal is the algebraic sum of the motor unit action potentials (MUAP) generated by the active motor units (MU) and detected over the skin. Like any other EMG signal, sEMG provides quantitative information concerning wave-shape, amplitude, power spectral density, etc. We assumed that the signals are the consequence of muscle activation over the entire exercise proposed to the horse.

            Line 376: I’m still really struggling to follow this description. What is “burst like muscular activity”? A single burst? Multiple bursts? Do these occur during ipsilateral swing phase during the forward motion? Where does the reported swing phase occur during the destabilisation exercise? Needs to be more clearly described. No activation bursts were detected during the backwards motion? Was only baseline activity detected? The authors state that they detected onset and activation duration, but this is not included in the descriptions for any of the muscles?

Author response: Thank you for the comments. The term “burst like muscular activity” refer to the signal visualized in the activation map; this is a colorimetric elaboration of an electrical event describing a small frame of an entire exercise. This is only descriptive. The reader must bear in mind that the dataset comprising the original monopolar raw signals had an incredible amount of information in the range of milliseconds, because the acquisitions performed were around 10-20 seconds of activity. Therefore, description of activity of the muscle and activation presented in the figures are only representative of a small epoch and they show the analysis of the spatial distribution on HD-sEMG intensity. Concerning the timing of the swing phase and stance phase during a step forward and backward, horses were not instrumented with IMU sensors or registered with optical motion capture system during these exercises. Therefore, we are not in the position to provide this type of information synchronized with the electromyographic signals. We assumed that the activation of the SDP occurs during the swing phase of the forward destabilization of the horse. We considered that the HD-sEMG is not the best method to demonstrated muscle activation synchronized with movement; for this type of analysis bipolar EMG is probably better. HDsEMG provides a full range of spatial information that is impossible to capture with other EMG methods. To improve comprehension of data presented, data records presented in this section and accompanying detailed description file are now added as supplementary files. The records contain the raw signals without any further processing. Data are stored in individual folders for each of the subject. Files are named according to the type of task recorded.

            Figures 2 – 4 are very small and contain a lot of detail, so difficult to decipher, particularly the sEMG traces. Hopefully these will be much larger in the manuscript or could the authors consider presenting only the highlighted sections and then presenting figures with full traces a supplemental file?

Author response: Thank you. The figures were reformatted and presented in a way that are easily to decipher the sEMG signals.

            Greater attention to detail is required for figure legends:

Figure 3 does not reference the subpanel b. In figure 3, based on the sEMG traces, where does the exercise begin and end? Noise is apparent in some signals especially on the right side? Was this particular epoch chosen specifically to avoid this? Figure 4 description must be reviewed – it has quite a few errors for example the reference to c is an image and this description is repeated incorrectly for colour maps (line 432).

Figure 5 – again, please review the description for clarity. What does three mastication refer to? Trials? Cycles? N 1 – 3 labelling also unclear. I understand that this is one representative horse, but this should be clarified for the reader.

Author response: Thank you for the attentive review of the figures. Figures are now improved to better describe what they refer to. Signals where noise were apparent had been removed.

            Line 459: across all motor tasks? Including mastication or just the dynamic mobilization exercises studied? Please clarify.

Author response: The HDsEMG spectrum across all motor tasks were analyzed for fatigue.

            Figure 7: one representative horse? The description for figure 8 should include the same detail as figure 7, but it’s a little unclear as to what has been deleted in terms of this figure, as the tracked changes appear to distort this area of the document.

Author response: One representative horse is shown in Figure 7 and 8. The legend is now reviewed.

            Line 490: across all horses?

Can the authors consider including some tabular data for some of their variables? At the moment, the results presented are from individual subjects and are thus highly descriptive and subjective in nature. The ability to present some data from the entire dataset would greatly improve the work and conclusions drawn.

Author response: Data records presented in this section and accompanying detailed description file are now added as supplementary files. The records contain the raw signals without any further processing. Data are stored in individual folders for each of the subject. Files are named according to the type of task recorded. The reviewer must bear in mind that the dataset comprises the original monopolar raw signals had an incredible amount of information in the range of milliseconds and the acquisitions performed were around 10-20 seconds of activity. HD-sEMG data are difficult to be presented in aggregated form because the different duration of the motor tasks in term of time and the different percentage of maximal voluntary contraction (MVC). Only an isometric contraction of the same muscle in different animals performed at the same percentage of the MVC could be presented in a grouped way, but this is not the case.

            The discussion requires an extensive review of grammar, English language and sentence structure.

Author response: Done

Reviewer 4 Report

The paper was improved. However still some changes for clarification is needed.

  1. please add “ - preliminary result” to the title at the end of it
  2. please move the table with the characteristics (not demography) of horses from the results part into the material and methods part – to point 2.1
  3. the justification of treated masseter muscle as the control should be written wider in the method part (in discussion it is better described, but it should be clear from methods)
  4. reward (carrot?) mention in the text should be clear not only from the table description – from the methods
  5. please add into the point 2.4 that task were treated once left and right (?) and each lasted about … sec/min this info should be clear – in the figure 7 it is written – two repetitions of lateral flexion right? That must be good written everywhere – otherwise readers would not be able to understand what exactly was done.

It is also not clear where are the data from all horses presented.  Be clear the about regressions that are calculated – are they between electrodes or between all measurements for all horses? I do not see comparison of data between horses (fig. 2,3,4  – single horse; fig.5 –three chewings/horses?; fig.6 single horse; 7 –two repetitions –one horse?  Please specify this aspect. It should be clearly stated in the results and discussion. You investigated 10 horses but it is not written how were their data compared and what was the result of it. If it is a preliminary report it can be reported on low amount of data, but it should be clear.

  1. be sure that all figures have the description – I do not see now for example for the figure 2- everything seems withdrawn .

Author Response

The paper was improved. However still some changes for clarification is needed.

  1. please add “ - preliminary result” to the title at the end of it

Author response: Done

  1. please move the table with the characteristics (not demography) of horses from the results part into the material and methods part – to point 2.1

Author response: Done

  1. the justification of treated masseter muscle as the control should be written wider in the method part (in discussion it is better described, but it should be clear from methods)

Author response: In the method section a sentence better explaining this aspect is now added.

“A further muscle, the masseter muscle, were instrumented during spontaneous chewing, because activation of masticatory muscles is certainly obtained offering to the horse a reward and easily recorded due to the superficial position. The masseter muscle was considered in this study a positive control to demonstrate the feasibility of the system in horse.”

  1. reward (carrot?) mention in the text should be clear not only from the table description – from the methods

Author response: The use of reward to obtain motivation of the horses has better explained. A carrot was used to motivate horses for dynamic mobilization exercise. Hay was employed to obtain prolonged mastication.

  1. please add into the point 2.4 that task were treated once left and right (?) and each lasted about … sec/min this info should be clear – in the figure 7 it is written – two repetitions of lateral flexion right? That must be good written everywhere – otherwise readers would not be able to understand what exactly was done.

Author response: Thanks, this point is interesting. We do not specify this data in the point 2.4 because is highly subjective. We try to better explain this aspect with the following sentence in the point 2.4

Each motor task was repeated until a clean electromyography signal was recorded without mute channels recording or skin-to-electrodes motion artifacts. This mean that a coherent appearance of the myographic signals need to be detected by the examined during motor tasks recording before to acquire the record.”

In figure 7 two repetitions of the same motor task are shown to demonstrate how the sEMG signal modify its spectrum due to the onset of myoelectrical fatigue

            It is also not clear where are the data from all horses presented.  Be clear the about regressions that are calculated – are they between electrodes or between all measurements for all horses? I do not see comparison of data between horses (fig. 2,3,4  – single horse; fig.5 –three chewings/horses?; fig.6 single horse; 7 –two repetitions –one horse?  Please specify this aspect. It should be clearly stated in the results and discussion. You investigated 10 horses but it is not written how were their data compared and what was the result of it. If it is a preliminary report it can be reported on low amount of data, but it should be clear.

Author response: Thanks for the request of clarification. Each single regression is calculated and presented for a single motor task (one animal). This is because the entire spectrum of the sEMG activity is analyze. The graph represents the modification of the electromyographic spectrum during a sustained muscle contraction in a single horse. This is the method preferred also in human to show high-density surface electromyographic signal during muscle contraction, due to the large number of data recorder in a single session of work in the range of milliseconds by 64 sEMG channels acquiring simultaneously. (Rojas-Martinez M. et al. Scientific Data 2020 7:397/ https://doi.org/10.10387s41597-020-00717-6)

No comparison between horses has been done. fig. 2,3,4 – single horse yes; fig.5 –three chewings/horses? yes; fig.6 single horse yes; 7 –two repetitions –one horse? yes. this is better explained in the manuscript that it is a preliminary report describing this technology and its implication in horse

Data records presented in the manuscript and accompanying detailed description file are now added as supplementary files. The records contain the raw signals without any further processing. Data are stored in individual folders for each of the subject. Files are named according to the type of task recorded.

            be sure that all figures have the description – I do not see now for example for the figure 2- everything seems withdrawn .

Author response: The legend is now be added.